# HDTree: Generative Modeling of Cellular Hierarchies for Robust Lineage Inference

**Zelin Zang** [1 2 3] **WenZhe Li** [2] **Yongjie Xu** [2] **Chang Yu** [2] **Changxi Chi** [2] **Jingbo Zhou** [2] **Zhen Lei** [1 4 5] **Stan Z. Li** [2]

## Abstract

In single-cell research, tracing and analyzing high-throughput single-cell differentiation trajectories is crucial for understanding biological processes. Key to this is the robust modeling of hierarchical structures that govern cellular development. Traditional methods face limitations in computational cost, performance, and stability. VAE-based approaches have made strides but still require branch-specific network modules, limiting their scalability and stability, while often suffering from posterior collapse. To overcome these challenges, we introduce HDTree, a generative modeling framework designed for robust lineage inference. HDTree captures tree relationships within a hierarchical latent space using a unified hierarchical codebook and employs a quantized diffusion process to model continuous cell state transitions. By aligning the generative process with the Waddington landscape, this method not only improves stability and scalability but also enhances the biological plausibility of inferred lineages. HDTree's effectiveness is demonstrated through comparisons on both general-purpose and single-cell datasets, where it outperforms existing methods in lineage inference accuracy, reconstruction quality, and hierarchical consistency. These contributions enable accurate and efficient modeling of cellular differentiation paths, offering reliable insights for biological discovery.[1]

---

[1]Centre for Artificial Intelligence and Robotics (CAIR), Hong Kong Institute of Science and Innovation (HKISI), Hong Kong, China [2]School of Engineering, Westlake University, Hangzhou, Zhejiang 310030, China [3]Tsientang Institute for Advanced Study, Hangzhou, Zhejiang 310030, China [4]MAIS, Institute of Automation, Chinese Academy of Sciences (CASIA), Beijing, China [5]School of Artificial Intelligence, University of Chinese Academy of Sciences (UCAS), Beijing, China. Correspondence to: Zhen Lei <zhen.lei@ia.ac.cn>, Stan Z. Li <stan.z.li@westlake.edu.cn>.

*Proceedings of the 43rd International Conference on Machine Learning*, Seoul, South Korea. PMLR 306, 2026. Copyright 2026 by the author(s).

[1]Code is available at https://github.com/zangzelin/code_HDTree_icml.

## 1. Introduction

In single-cell research, tracing and analyzing cellular differentiation trajectories is essential for understanding dynamic biological processes. This task requires not only effective modeling of hierarchical structures (Zeng et al., 2022), but also the ability to generate data that faithfully captures such hierarchies (Guo et al., 2024). Accurately characterizing the hierarchical organization underlying cell differentiation facilitates the exploration of cellular systems and fate decisions, while conditional generation based on these hierarchies enables interpretable discovery of biological mechanisms. Hierarchical structures are ubiquitous in real-world data (Chehreghani & Chehreghani, 2024; Gyurek et al., 2024; Tian et al., 2024), yet modeling them remains challenging.

As shown in Fig. 1, traditional methods (Murtagh & Legendre, 2014) often rely on a combination of dimension reduction (Jia et al., 2022), clustering (Oti & Olusola, 2024), and data regression (Ali & Younas, 2021) techniques to achieve hierarchical modeling and data generation. While these approaches can address the tasks to some extent, they face challenges regarding computation costs, performance, generative capacity, and stability (Zang et al., 2025c). These limitations make them inadequate for handling the demands of large-scale, high-dimensional biological data. VAE-based methods (Manduchi et al., 2023; Xiao & Su, 2024; Majima et al., 2024) unify generation and representation, reducing costs compared to traditional approaches. However, a key limitation of existing SOTA methods lies in their reliance on specialized network modules for each tree branch (Manduchi et al., 2023). This design not only reduces stability but also constrains the ability to capture deep hierarchies. Specifically, deep branches with sparse samples cannot effectively leverage representation knowledge learned from other branches, leading to limited generalization and difficulty in preserving global structure (Ghahramani et al., 2010; Lakshminarayanan et al., 2016). Moreover, independent encoder-decoder pairs at each branch node are prone to overfitting due to noise accumulation when training data is limited, thereby limiting applicability in scenarios requiring robust and deep hierarchical modeling.

To address these challenges, we propose **H**ierarchical vec-

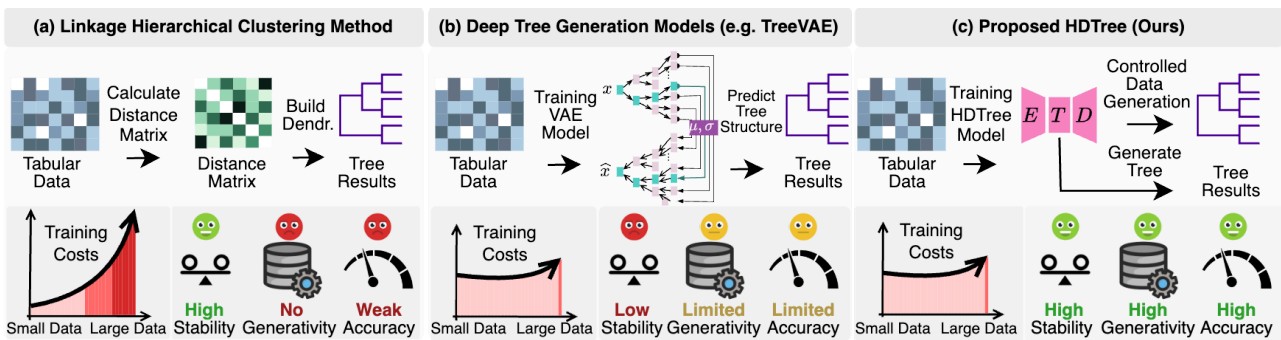

Figure 1. **Motivation.** Existing shallow and deep hierarchical methods cannot fully meet the requirements of hierarchical representation and lineage analysis in terms of stability, generativity, accuracy, and training cost.

tor quantized **D**iffusion Model (**HDTree**), which captures tree relationships in a hierarchical latent space through a *unified hierarchical codebook* (Huang et al., 2024) and models branch transitions via a quantized diffusion process (Gu et al., 2022). The core innovation of HDTree lies in its integration of hierarchical latent space encoding with a quantized diffusion process, systematically addressing the aforementioned limitations. First, enhanced generalization is achieved by employing a unified latent space where all branches share the same codebook vectors, enabling even sparse deep nodes to leverage representation knowledge from other branches while preserving global hierarchical structure. Second, improved stability is ensured by replacing branch-specific encoder-decoder pairs with a unified encoder and hierarchical codebook architecture, reducing noise accumulation and overfitting risks associated with independent modules while maintaining adaptability to complex tree topologies. Third, strengthened generative capacity is realized by modeling branch transitions via a diffusion process (Liu et al., 2024). Unlike VAEs' single-step generation which risks posterior collapse, diffusion's iterative denoising naturally mirrors the Waddington landscape—modeling differentiation as a gradual refinement from coarse progenitors to fine-grained specialized states. Finally, performance gains arise from soft contrastive learning and multi-scale latent space regularization, which sharpen the representation of hierarchical dependencies and improve lineage analysis accuracy.

**Our contributions are threefold: Methodological Innovation:** We introduce HDTree, a unified framework that integrates hierarchical vector quantization with diffusion processes. Unlike prior works that rely on fragmented, branch-specific modules, HDTree utilizes a single, shared hierarchical codebook. This design effectively decouples tree complexity from network size, ensuring scalability to deep biological hierarchies. **Application to Lineage Dynamics:** We propose a lineage inference module that leverages the learned hierarchical latent space. By perform-

ing shortest-path analysis over the learned tree-conditioned graph, HDTree recovers robust developmental trajectories even in sparse data regimes. **Empirical Superiority:** Extensive evaluation across general benchmarks and complex single-cell datasets demonstrates that HDTree consistently outperforms state-of-the-art baselines (including TreeVAE and foundation models) in reconstruction quality, clustering accuracy, and lineage consistency.

**Conflict of Interest Disclosure.** The authors declare no financial conflicts of interest related to this work.

## 2. Related Work

**Tree-Structured Representation & Generative Models** Tree-structured representations are crucial for modeling hierarchical relationships in data (Zang et al., 2025c). Traditional methods like hierarchical clustering (Müllner, 2011) and distance-based techniques (Bouguettaya et al., 2015) rely on predefined metrics for tree construction. Recent deep learning advancements, such as TreeVAE (Manduchi et al., 2023), leverage recursive and hierarchical latent structures. Hyperbolic geometry methods, including HGNNs (Zhou et al., 2023), offer efficient hierarchical representations. In single-cell analysis, deep manifold learning methods preserve geometric or hierarchical structure for visualization and trajectory-related tasks (Xu et al., 2023; 2025; Zang et al., 2025a). TreeVI (Xiao & Su, 2024) enhances variational inference by utilizing tree structures for scalable training and improved performance in tasks like clustering and link prediction. **Deep Learning Based Cell Lineage Analysis** Cell lineage analysis is crucial for reconstructing developmental trajectories in single-cell genomics. Traditional methods like *Monocle* (Trapnell et al., 2014) and *Slingshot* (Street et al., 2018) infer pseudotime trajectories but are limited by predefined metrics and difficulty modeling unobserved progenitor states. Recent approaches such as *LineageVAE* (Majima et al., 2024), *Waddington-OT* (Schiebinger et al., 2019), and neural transport mod-

els (Chi et al., 2026) overcome some limitations with probabilistic models and optimal transport, though high dimensionality and sparsity remain challenges. Detailed description of the related work is provided in Appendix.

# 3. Methods

## 3.1 Notation and Task Definition

Let $\mathcal{X} = \{\mathbf{x}_i \in \mathbb{R}^D\}_{i=1}^N$ denote a dataset with $N$ samples, where each $\mathbf{x}_i$ is a $D$-dimensional feature vector. To enhance the generalization capability and robustness of the model, an augmented view $\mathbf{x}_i^+$ is generated for each $\mathbf{x}_i$ using kNN-based augmentation (Zang et al., 2025b): we sample $\mathbf{x}_i^+$ from the local neighbor set of $\mathbf{x}_i$, encouraging the encoder to preserve local semantic neighborhoods during contrastive learning. HDTree learns a hierarchical tree-structured latent representation $\mathcal{T}$ to capture multi-scale semantic relationships among data points. Formally, $\mathcal{T}$ is parameterized as a rooted binary tree of maximum depth $L$, where each node at depth $l$ is associated with a learnable code vector $\mathbf{w}_j^l \in \mathbb{R}^d$. Both the node embeddings and the tree routing are jointly optimized during training, allowing the model to automatically discover a data-driven hierarchical organization.

**Task 1 (Lineage Analysis).** Given the learned hierarchical tree $\mathcal{T}$, the lineage analysis task aims to infer developmental trajectories by identifying paths that connect a specified origin node (e.g., a progenitor or stem cell state) to one or more destination nodes (e.g., differentiated cell types). The resulting paths represent discrete approximations of cell state transitions and reveal the hierarchical progression of differentiation.

**Task 2 (Generative Validation via Lineage Conditioning).** Beyond trajectory inference, HDTree leverages its generative capability to validate the learned hierarchy. By synthesizing new data points conditioned on specific lineage paths, the diffusion decoder $\mathcal{D}_\theta$ simulates intermediate or hypothetical cell states. This "in-silico" experimentation ensures that the inferred trajectories correspond to biologically plausible transitions on the data manifold.

## 3.2 Model Design

HDTree addresses three key challenges in deep hierarchical modeling: (1) scalable representation of deep trees with exponentially growing branches, (2) capturing multi-granularity relationships across hierarchical levels, and (3) generating diverse samples along specific biological paths. These requirements motivate our design of three synergistic modules: HDTree is achieved through three key modules: encoder $\mathcal{E}_{\theta_E}$, hierarchical VQ codebook $\mathcal{C}_W$, and diffusion decoder $\mathcal{D}_\theta$.

**Encoder $\mathcal{E}_{\theta_E}$ & Hierarchical Tree Codebook (HTC) $\mathcal{C}_W$.** Unlike prior methods where network parameters scale exponentially with tree depth (Manduchi et al., 2023), HDTree avoids the exponential growth of network parameters. The hierarchical structure is explicitly encoded through a unified codebook, decoupling the tree complexity from the neural network architecture. The encoder $\mathcal{E}_{\theta_E}$ maps input data $\mathbf{x}_i$ into a latent space $\mathbf{z}_i \in \mathbb{R}^d$, where $\theta_E$ represents the learnable parameters of the encoder and $d$ is the latent dimensions. The encoding process can be expressed as, $\mathbf{z}_i = \mathcal{E}_{\theta_E}(\mathbf{x}_i), \mathbf{z}_i^+ = \mathcal{E}_{\theta_E}(\mathbf{x}_i^+)$.

To capture the hierarchical relationships in the data, HTC $\mathcal{C}_W$ is introduced, which is constructed as a binary tree, where each node represents a code vector in the latent space,

$$\mathcal{T}_j^l = \mathbf{w}_j^l \text{ if } l = L, \quad (\mathbf{w}_j^l, \mathcal{T}_{2j}^{(l+1)}, \mathcal{T}_{2j+1}^{(l+1)}) \text{ if } l < L. \quad (1)$$

where $\mathbf{w}_j^l$ is the learnable code vector at depth $l$ and index $j$, $\mathcal{T}_0^0$ is the root node of the tree, and $L$ is the maximum depth of the tree. The $\mathbf{w}_j^l \in \mathbb{R}^d$ denotes the node embedding at level $l$; for a full binary scaffold, level $l$ contains up to $2^l$ nodes. The tree structure is optimized during training to capture the semantic and structural relationships in the data. For input data $\mathbf{x}_i$, the HTC is used to quantize the latent representation $\mathbf{z}_i$ into a hierarchical sequence of code vectors $\mathbf{s}_i$,

$$\mathbf{s}_i = [\Omega^{(1)}(\mathbf{z}_i), \dots, \Omega^{(l)}(\mathbf{z}_i), \dots, \Omega^{(L)}(\mathbf{z}_i)],$$
$$\Omega^{(l)}(\mathbf{z}_i) = \operatorname{argmin}_{\mathbf{w}_j^l \in \text{Children}(\Omega^{(l-1)}(\mathbf{z}_i))} \|\mathbf{z}_i - \mathbf{w}_j^l\|_2, \quad (2)$$

where $\Omega^{(l)}(\mathbf{z}_i)$ selects the nearest code vector at depth $l$, and $\Omega^{(0)}(\mathbf{z}_i)$ denotes the root code. In the rare case of a tie (i.e., multiple $\mathbf{w}_j^l$ with identical distance), we break ties deterministically by choosing the codeword with the smallest index $j$, ensuring that $\mathbf{s}_i$ is uniquely defined. The Children$(\Omega^{(l-1)}(\mathbf{z}_i))$ denotes the children of the previously selected code vector at level $l-1$, as defined in Eq. (1). This design ensures sibling nodes (e.g., $\mathbf{w}_{2j}^{l+1}$, $\mathbf{w}_{2j+1}^{l+1}$) naturally inherit and refine their parent code $\mathbf{w}_j^l$, enabling knowledge sharing across branches.

**Diffusion Decoder $\mathcal{D}_{\theta_D}$** Unlike VAE decoders that suffer from posterior collapse and rely on single-step generation, and aligning with the Waddington landscape where differentiation is a continuous refinement, the diffusion decoder $\mathcal{D}_{\theta_D}$ in HDTree explicitly aligns the generation process to the hierarchical codebook through quantized conditioning. It reconstructs data or generates new samples based on the hierarchical latent representations, encouraging valid path traversal across tree levels. It leverages a Denoising Diffusion Probabilistic Model (DDPM) (Ho et al., 2020) to iteratively generate data starting from a noise distribution. Specifically, beginning with Gaussian noise $\mathbf{x}_T \sim \mathcal{N}(0, \mathbf{I})$, the model refines $\mathbf{x}_T$ through $T$ diffusion steps to produce the final data $\mathbf{x}_0$ conditioned on the quantized code sequence $\mathbf{s}_i$ obtained from VQ (Eq. 5). The generation process

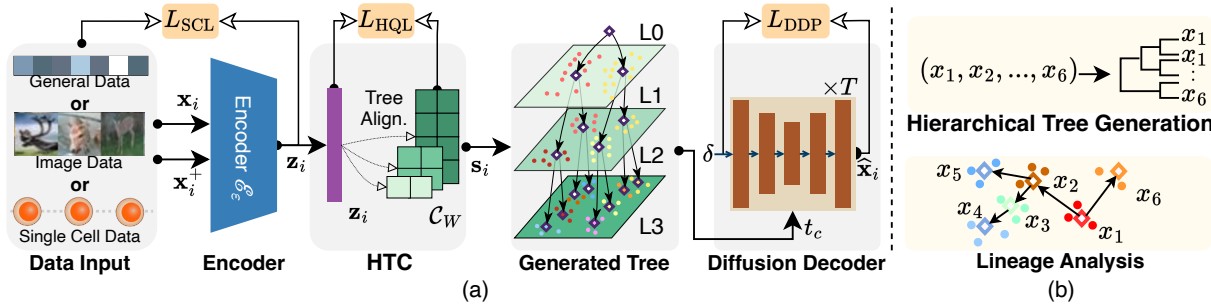

*Figure 2.* **Overview of the HDTree framework & tasks.** (a) The framework of HDTree, which consists of three main components: encoder for semantic representation, Hierarchical Tree Codebook (HTC) for tree-based structural modeling, and diffusion-based decoder for generative validation. We use tree structures to model hierarchical relationships and validate the manifold continuity based on the hierarchical latent space. The soft contrastive loss (SCL), hierarchical quantization loss (HQL), and diffusion loss (DDP) are used to optimize the model. (b) The hierarchical representation learning and lineage inference.

$\widetilde{\mathbf{x}}_i = \mathrm{Gen}(\delta, \mathbf{s}_i | \mathcal{D}_{\theta_D}(\cdot))$ is formulated as,

$$\mathrm{Gen}(\delta, \mathbf{s}_i | \phi^*) = \left\{ \widetilde{\mathbf{x}}^0 \,\Big|\, \widetilde{\mathbf{x}}^{t-1} = \frac{1}{\sqrt{\alpha_t}} \left( \widetilde{\mathbf{x}}^t - \widetilde{\alpha} \right) + \sigma_t \epsilon_t \right\},$$

$$\widetilde{\alpha} = \frac{1 - \alpha_t}{\sqrt{1 - \bar{\alpha}_t}} \mathcal{D}_{\theta_D}(\widetilde{\mathbf{x}}^t, t, \mathbf{s}_i), \tag{3}$$

where $t \in \{T, \cdots, 1\}$, $\epsilon_t \sim \mathcal{N}(0, \mathbf{I})$, $\mathbf{s}_i = \{c_{\mathbf{z}_i}^1, \ldots, c_{\mathbf{z}_i}^L\}$ is the hierarchical code sequence from root to leaf, and $\mathcal{D}_{\theta_D}(\cdot)$ is a neural network approximator that predicts noise $\delta$ conditioned on both the noisy sample $\widetilde{\mathbf{x}}^t$ and the hierarchical path $\mathbf{s}_i$, ensuring generated samples conform to the learned tree structure. Although the hierarchical codebook $\mathcal{C}_W$ is parameterized as a full binary tree for efficient indexing, this does not restrict the induced hierarchy to a strictly binary interpretation. Each data point follows a binary latent path during quantization, but multiple points can share partial paths and diverge at different levels. When these paths are aggregated over samples, they naturally induce multi-branch groupings and unbalanced hierarchical structures, with the binary tree serving as an indexing mechanism for hierarchical codes.

### 3.3 Loss Function Design

We optimize HDTree using a composite loss integrating contrastive learning, vector quantization, and diffusion-based reconstruction.

**Soft Contrastive Learning Loss (SCL) [$\mathcal{L}_{\mathbf{SCL}}(\cdot)$]** To preserve hierarchical granularity, we adapt SCL (Zang et al., 2023; 2025b) to weight negative pairs by tree distance. For a batch of embeddings $\mathbf{z} = \{\mathbf{z}_i\}_{i=1}^{N_b}$ and augmentations $\{\mathbf{z}_i^+\}_{i=1}^{N_b}$, the loss is $\mathcal{L}_{\mathrm{SCL}} =$

$$\frac{1}{2N} \sum_{i_1=1}^{N_b} \left( \log \sum_{i_2=1}^{N_b} \mathbf{S}_{i_1 i_2}^{\mathbf{z}\mathbf{z}^+} + \log \sum_{i_2=1}^{N_b} \mathbf{S}_{i_1 i_2}^{\mathbf{z}^+\mathbf{z}} \right) - \\ \sum_{i_1=1}^{N_b} \log \mathrm{diag}(\mathbf{S}_{i_1 i_1}^{\mathbf{z}\mathbf{z}^+}), \tag{4}$$

where $N_b$ is the batch size, $\mathbf{S}_{ij}^{\mathbf{z}\mathbf{z}^+}$ represents the similarity

matrix calculated using the t-distribution kernel, $\mathbf{S}_{i_1 i_2} = \left(1 + (\mathbf{D}_{i_1 i_2}^2)/\nu\right)^{-\frac{\nu+1}{2}}$, and $\mathbf{D}_{i_1 i_2}$ is the pairwise distance between $\mathbf{z}_{i_1}$ and $\mathbf{z}_{i_2}$ in the hyperbolic space, where $\nu = 0.1$ is the degrees of freedom of the t-distribution.

**Hierarchical Quantization Loss (HQL) [$\mathcal{L}_{\mathbf{HQL}}(\cdot)$]** HQL learns robust hierarchical tree-structured representations in the HTC by aligning latent embeddings with multi-level code vectors while maintaining inter-level consistency. The loss is defined as,

$$\mathcal{L}_{\mathrm{HQL}} = \sum_{l=1}^{L} \mathcal{A}(\mathbf{z}_i, c_{\mathbf{z}_i}^l) + \lambda \mathcal{A}(c_{\mathbf{z}_i}^l, \Psi^{\mathbf{z}_i}(c_{\mathbf{z}_i}^l)), \tag{5}$$

where $\Psi^{\mathbf{z}}(\mathbf{w}_j) = \arg\min_{\mathbf{z}_i \in \mathbf{z}} \|\mathbf{z}_i - \mathbf{w}_j\|_2$, $c_{\mathbf{z}_i}^l$ is the nearest code vector to $\mathbf{z}_i$ at level $l$, and $\lambda = 2$ balances alignment and consistency. The first term $\mathcal{A}(\mathbf{z}_i, c_{\mathbf{z}_i}^l)$ aligns embeddings with codes to preserve parent-child relations, while the second term $\mathcal{A}(c_{\mathbf{z}_i}^l, \Psi^{\mathbf{z}_i}(c_{\mathbf{z}_i}^l))$ enforces consistency by mapping codes back to their nearest embeddings, separating sibling nodes and anchoring children to their parents to maintain a coherent hierarchy. The $\mathbf{z} = \{\mathbf{z}_i\}_{i=1}^{N_b}$ denotes the latent embeddings in the current batch. The function $\mathcal{A}(a, b) = \|\mathbf{sg}(a) - b\|_2^2 + \|a - \mathbf{sg}(b)\|_2^2$, where $\mathbf{sg}(\cdot)$ denotes the stop-gradient operation.

**Diffusion Loss [$\mathcal{L}_{\mathbf{DDP}}(\cdot)$]** We optimize the standard variational lower bound (ELBO) as in Ho et al. (Ho et al., 2020).

$$\mathcal{L}_{\mathrm{DDP}} = \mathbb{E}_{t \sim [1, T], \mathbf{x}, \epsilon \sim \mathcal{N}(0, I)} \left[ \left\| \epsilon - \epsilon_{\theta_D}(\sqrt{\bar{\alpha}_t}\mathbf{x} + \sqrt{1 - \bar{\alpha}_t}\epsilon, t, \mathbf{s}_i) \right\|_2^2 \right], \tag{6}$$

where $\epsilon_{\theta_D}(\cdot)$ is the predicted Gaussian noise, $\beta_t$ is the variance schedule, $\alpha_t = 1 - \beta_t$, and $\bar{\alpha}_t = \prod_{s=1}^{t} \alpha_s$ is the cumulative product term. The conditional vector $\mathbf{s}_i$ is obtained from Eq. (2). This loss enforces the decoder to match the true noise $\epsilon$ and thus ensures faithful reconstruction of $\mathbf{x}$ while respecting the hierarchical conditions.

**Overall Loss Function.** The overall loss is defined as,

$$\mathcal{L} = \mathcal{L}_{\mathrm{SCL}} + \lambda_{\mathrm{HQL}}\mathcal{L}_{\mathrm{HQL}} + \lambda_{\mathrm{DDP}}\mathcal{L}_{\mathrm{DDP}}, \tag{7}$$

where $\lambda_{\text{HQL}}$, and $\lambda_{\text{DDP}}$ are hyperparameters controlling the contributions of the contrastive learning loss, vector quantization loss, and diffusion-based reconstruction loss, respectively. This composite loss ensures balanced optimization across all components of the HDTree model.

### 3.4 Trajectory Analysis with HDTree

**Graph Construction.** To infer developmental trajectories, the hierarchical tree structure generated by HDTree is transformed into a weighted graph $\mathcal{G} = (\mathcal{V}, \mathcal{E}, \mathcal{W})$. The graph $\mathcal{G}$ inherits all nodes and edges from the hierarchical tree $\mathcal{T}$, while additional edges are added within the same depth using a $k$-nearest neighbors (KNN) approach. This augmentation enriches the connectivity of the graph by capturing local semantic relationships that are not explicitly represented in the original tree structure. The nodes in the graph correspond to the code vectors $\mathbf{w}_j^l$ at each depth $l$ and index $j$. The edges include both the hierarchical edges from $\mathcal{T}$, denoted by $\mathcal{E}_{\mathcal{T}}$, and the newly introduced edges generated by the KNN process, denoted by $\mathcal{E}_{\text{KNN}}$. For each edge $(j_1, j_2)$ in the graph, the weight is,

$$\mathbf{w}(j_1, j_2) = \begin{cases} \|\mathbf{w}_{j_1} - \mathbf{w}_{j_2}\|_2, & (j_1, j_2) \in \mathcal{E}_{\mathcal{T}}, \\ \|\mathbf{w}_{j_1} - \mathbf{w}_{j_2}\|_2 + P^{L-l}, & (j_1, j_2) \in \mathcal{E}_{\text{KNN}}, \end{cases}$$
$$(8)$$

where $l$ denotes the depth of the auxiliary KNN edge and $P^{L-l}$ is a depth-dependent penalty, encouraging developmental trajectories to follow the hierarchical structure while still allowing local lateral connections when supported by the learned representation.

**Trajectory Inference.** Using the constructed graph $\mathcal{G}$, we infer developmental trajectories by finding the shortest path between a designated origin $\mathbf{w}_{\text{start}}$ and destination $\mathbf{w}_{\text{end}}$. The shortest path is determined by minimizing the total edge weights along the trajectory:

$$\text{Path}_{\text{development}} = \underset{\text{Path} \subseteq \mathcal{G}}{\arg\min} \sum_{(j_1, j_2) \in \text{Path}} \mathbf{w}(j_1, j_2). \quad (9)$$

The inferred developmental trajectories comprehensively represent the underlying hierarchical relationships in the data, capturing the transitions between different cell states and the progression of cell differentiation. The KNN connections serve only as an auxiliary augmentation to enhance local connectivity and do not alter the global hierarchy encoded by $\mathcal{T}$. Empirically, our results are robust across a wide range of $k$ values (see Appendix for sensitivity analysis), indicating that the primary performance gain comes from the tree-structured representation itself.

## 4. Experiments

**Datasets & Baseline Methods.** We evaluate HDTree on four general datasets (MNIST, Fashion-MNIST, 20News-Groups, CIFAR-10) and four single-cell datasets (Limb(Zhang et al., 2024), LHCO(He et al., 2022), Weinreb(Weinreb et al., 2020), ECL(Qiu et al., 2024)). We compare HDTree against traditional clustering methods, VAE-based approaches, and single-cell specific models (in Table 1 and 2 details in Appendix).

**Evaluation Metrics.** To comprehensively evaluate HDTree and baseline methods, the testing protocol is divided into three parts: clustering performance, tree structure performance, and reconstruction performance. *Clustering performance* is measured using *Clustering Accuracy (ACC) (Nazeer et al., 2009)* and *Normalized Mutual Information (NMI)(Estévez et al., 2009)*. The input data is first mapped into a latent space using the respective method. The clustering results are then derived directly from this latent representation. For methods that do not inherently produce clustering results, hierarchical clustering is applied to the latent space to generate cluster labels. Tree structure performance (Tree performance) is evaluated using *Leaf Purity (LP) (Manning et al., 2008)* and *Dendrogram Purity (DP)(Rokach & Maimon, 2005)*. We predict the tree structures with different methods. **Reconstruction performance** is assessed using *Reconstruction Loss (RL)* and *Log-Likelihood (LL)*. These metrics quantify the ability of a method to recover the original input data from its latent space representation.

**Testing Protocol & Implementation.** Detailed hyperparameters and experimental setup are provided in Appendix.

**Comparisons on General Datasets [better stability/accuracy/generativity].** The proposed HDTree is a hierarchical representation learning and generative modeling framework. To ensure a fair comparison of clustering, tree construction, and generation performance across different methods, we adopted the benchmarking strategy described in the benchmark of (Manduchi et al., 2023). The results are shown in Table 1. The symbol [A] means the methods directly use the agglomerative clustering method on the embeddings to calculate the Tree Performance. **Analysis:** (1) HDTree achieves superior performance across all metrics compared to traditional and SOTA methods, with advantages becoming more pronounced as dataset complexity increases. This robustness stems from its explicit modeling of hierarchical structures, which effectively captures complex data relationships. (2) Unlike TreeVAE and the agglomerative variant HDTree[A], HDTree utilizes a unified tree representation framework that ensures lower variance and enhanced stability. This structural advantage effectively boosts modeling and generation capabilities, validating the necessity of the proposed hierarchical mechanism over alternative approaches.

**Comparisons on Single Cell Datasets [better stability/accuracy].** Due to the lack of established benchmarks in the

*Table 1.* **Comparison of tree performance, clustering performance, and reconstruction performance (Rec. Performance) on four general image and text datasets.** The symbol [A] means directly use agglomerative clustering on the embeddings to get the tree performance. The -RL and LL are the reconstruction loss and negative log-likelihood. The best results are highlighted in **bold**. The number after/before $\pm$ shows the mean/standard deviation with 10 different random seeds. '-' indicates these methods do not have the generation ability.

| Dataset | Method | Tree Performance | | Clustering Performance | | Rec. Performance | | Average |
|---|---|---|---|---|---|---|---|---|
| | | DP(↑) | LP(↑) | ACC(↑) | NMI(↑) | -RL(↑) | LL(↑) | |
| Mnist (image,70k×784) | Agg | 63.7±0.0 | 78.6±0.0 | 69.5±0.0 | 71.1±0.0 | - | - | - |
| | VAE[A] | 79.9±2.2 | 90.8±1.4 | 86.6±4.9 | 81.6±2.0 | -84.7±2.6 | -87.2±2.0 | 27.8±2.5 |
| | LadderVAE[A] | 81.6±3.9 | 90.9±2.5 | 80.3±5.6 | 82.0±2.1 | -87.8±0.7 | -99.9±0.3 | 24.5±2.5 |
| | DeepECT | 74.6±5.9 | 90.7±3.2 | 74.9±6.2 | 76.7±4.2 | - | - | - |
| | TreeVAE | 87.9±4.9 | 96.0±1.9 | 90.2±7.5 | 90.0±4.6 | -80.3±0.2 | -92.9±0.2 | 31.8±3.2 |
| | HDTree[A] | **92.7±0.3** | **97.1±1.2** | **97.1±0.1** | **92.8±0.2** | - | - | - |
| | HDTree | 91.9±2.8 | 96.6±1.4 | 96.6±1.4 | 92.4±1.3 | **-77.9±1.2** | **-85.4±1.4** | **35.7±1.6 (↑3.9)** |
| Fashion-Mnist (image,70k×784) | Agg | 45.0±0.0 | 67.6±0.0 | 51.3±0.0 | 52.6±0.0 | - | - | - |
| | VAE[A] | 44.3±2.5 | 65.9±2.3 | 54.9±4.4 | 56.1±3.2 | -231±3.2 | -242±3.2 | -32.1±3.1 |
| | LadderVAE[A] | 49.5±2.3 | 67.6±1.2 | 55.9±3.0 | 60.7±1.4 | -231±1.4 | -239±1.4 | -39.5±1.8 |
| | DeepECT | 44.9±3.3 | 67.8±1.4 | 51.8±5.7 | 57.7±3.7 | - | - | - |
| | TreeVAE | 53.4±2.4 | 70.4±2.0 | 60.6±3.3 | 64.7±1.4 | -226±1.4 | -234±1.4 | -35.4±2.0 |
| | HDTree[A] | **47.7±1.6** | **67.1±1.5** | **64.6±1.9** | **67.4±1.2** | - | - | - |
| | HDTree | **57.4±0.3** | **71.8±0.3** | **71.1±0.2** | **68.7±0.2** | **-219±0.1** | **-228±0.1** | **-29.9±0.2 (↑5.5)** |
| 20news-groups (text,19k×2000) | Agg | 13.1±0.0 | 30.8±0.0 | 26.1±0.0 | 27.5±0.0 | - | - | - |
| | VAE[A] | 7.1±0.3 | 18.1±0.5 | 15.2±0.4 | 11.6±0.3 | -45.5±0.1 | -44.2±0.3 | -6.3±0.3 |
| | LadderVAE[A] | 9.0±0.2 | 20.0±0.7 | 17.4±0.9 | 17.8±0.6 | -43.5±0.1 | -44.3±0.6 | -3.9±0.5 |
| | DeepECT | 9.3±1.8 | 17.2±3.8 | 15.6±3.0 | 18.1±4.1 | - | - | - |
| | TreeVAE | 17.5±1.5 | 38.4±1.6 | 32.8±2.3 | 34.4±1.5 | -34.4±1.5 | -34.4±1.5 | 9.1±1.7 |
| | HDTree [A] | **22.0±0.1** | **45.5±0.4** | **44.6±0.4** | **43.7±0.2** | - | - | - |
| | HDTree | 23.7±0.1 | 44.0±0.2 | 41.8±0.2 | 42.6±0.2 | **-31.1±0.3** | **-34.1±1.5** | **19.0±0.4 (↑9.9)** |
| Cifar10 (image, 50k×32×32) | VAE[A] | 10.5±2.3 | 16.3±2.3 | 16.3±1.6 | 1.86±4.2 | **-31.7±2.9** | **-39.2±2.9** | -4.3±2.7 |
| | LadderVAE[A] | 12.8±3.9 | 25.3±3.9 | 25.3±2.0 | 7.41±4.9 | -41.8±4.7 | -40.2±3.7 | -1.9±3.9 |
| | DeepECT | 10.5±2.5 | 10.3±2.5 | 10.3±2.8 | 0.18±4.2 | - | - | - |
| | TreeVAE | 35.3±4.0 | 53.8±3.9 | 52.9±7.0 | 41.4±5.9 | -47.0±5.9 | -48.3±2.4 | 14.7±4.9 |
| | HDTree[A] | **44.2±1.5** | **55.2±1.8** | **75.9±4.3** | **55.3±2.5** | - | - | - |
| | HDTree | 43.8±1.7 | 55.1±1.4 | 73.2±2.7 | 53.9±2.0 | -34.7±1.9 | -40.3±3.6 | **25.2±2.2 (↑10.5)** |

single-cell domain, we incorporated data from high-impact published studies into the TreeVAE benchmark. The results are shown in Table 2. **Analysis:** (1) Similar to the general dataset, HDTree consistently demonstrates superior performance across all evaluated metrics, including tree structure quality, clustering accuracy, and hierarchical integrity. (2) **Zero-shot vs. Unsupervised Training:** We compare HDTree (trained unsupervised) against foundational single-cell models (Geneformer, CellPLM) in a zero-shot setting. While these settings are not directly comparable because HDTree is trained on the target data, this comparison underscores HDTree's core strength: efficient and domain-specific modeling without the need for pre-trained large model resources. Unlike foundational models that often require computationally expensive fine-tuning to capture fine-grained hierarchies, HDTree offers a lightweight and stable alternative for precise single-cell analysis. (3) These results establish HDTree as a robust and reliable approach for single-cell data analysis.

**Comparisons on Lineage Ground Truth [better stability/accuracy].** We evaluate latent alignment with developmental progression using the ratio of observed time points in the $k$-nearest neighborhood ($k = 30$). Table 3 shows HDTree consistently outperforms baselines. On LineageVAE data, HDTree achieves the highest temporal consistency, surpassing semi-supervised LineageVAE by +1.7% at Day 6, highlighting our unsupervised modeling advantage. **Analysis:** On *C. elegans*, HDTree maintains strong performance across all stages, improving over TreeVAE by > 4.0% in later windows and outperforming supervised LineageVAE early on. These results confirm HDTree's hierarchical structure faithfully reflects differentiation dynamics and generalizes robustly.

**Case Study on Generative Validation [validating lineage plausibility].** The generative capability of HDTree serves as a powerful validation tool for the inferred lineages. By simulating the transformation processes between different tree branches (e.g., from stem cells to somatic cells), we can

*Table 2.* **Comparison of tree performance, clustering performance on three single cell datasets.** Since most of the methods are not generative models, we did not compare generative performance.

| Dataset | Method | Year | Tree Performance | | Clustering Performance | | Average(↑) |
|---|---|---|---|---|---|---|---|
| | | | DP(↑) | LP(↑) | ACC(↑) | NMI(↑) | |
| Limb (cell lineage, 66,633 cells, celltype:10) | Geneformer[A] | 2023 | 25.6±5.4 | 35.9±0.1 | 34.1±0.1 | 34.9±0.1 | 32.6±1.4 |
| | CellPLM[A] | 2024 | 25.6±0.1 | 39.9±0.1 | 34.1±0.2 | 32.9±0.2 | 33.1±0.2 |
| | LangCell[A] | 2024 | 25.3±0.1 | 37.5±0.1 | 33.9±0.1 | 35.1±0.1 | 33.0±0.1 |
| | TreeVAE[A] | 2024 | 34.7±1.7 | 55.6±1.0 | 49.8±0.1 | 50.0±0.0 | 47.5±0.7 |
| | HDTree[A] | Ours | **38.9±1.3** | **57.9±1.0** | **52.8±1.0** | 49.0±0.1 | **49.7±0.9** |
| | HDTree | Ours | **41.0±0.4** | **57.2±1.4** | **55.0±1.4** | 46.6±0.4 | **50.0±0.9** (↑2.5) |
| LHCO (cell lineage, 10,628 cells, celltype:7) | CellPLM[A] | 2024 | 27.0±1.1 | 35.8±2.7 | 16.8±3.4 | 1.65±5.2 | 20.3±3.1 |
| | LangCell[A] | 2024 | 26.5±1.2 | 35.2±0.8 | 35.2±0.6 | 0.02±0.9 | 24.2±0.9 |
| | TreeVAE[A] | 2024 | 38.3±2.0 | 52.2±0.1 | 37.9±0.1 | 31.6±0.0 | 40.0±0.6 |
| | HDTree[A] | Ours | **38.8±0.3** | **52.1±0.4** | **46.4±0.3** | **34.7±0.5** | **43.0±0.4** |
| | HDTree | Ours | **42.7±0.4** | **54.0±0.3** | **49.4±0.3** | 34.5±0.4 | **45.2±0.3** (↑2.2) |
| Weinreb (cell lineage, 130,887 cells, celltype:11) | LangCell[A] | 2024 | 47.4±0.1 | 54.8±0.0 | 14.3±0.5 | 34.3±0.0 | 37.7±0.2 |
| | Geneformer[A] | 2024 | 45.1±0.4 | 55.3±0.1 | 21.4±0.1 | 32.3±0.1 | 38.5±0.2 |
| | TreeVAE[A] | 2024 | 60.4±2.6 | 61.4±0.5 | 41.0±0.1 | 35.2±0.0 | 49.5±0.8 |
| | HDTree[A] | Ours | **63.3±2.6** | **78.2±1.1** | **50.6±1.0** | **45.2±1.2** | **59.3±1.5** (↑7.5) |
| | HDTree | Ours | **61.0±0.4** | **67.0±0.3** | **62.6±0.3** | 42.6±0.3 | **58.3±0.4** |

*Table 3.* **Comparisons on Lineage Ground Truth.** The ratio of observed time points (Lineage Ground Truth) in the $k$-neighborhood ($k=30$). Top: LineageVAE dataset. Bottom: *C. elegans* dataset.

| | Time | Waddington OT | LineageVAE (semi-supervised) | scVI+Agg (unsupervised) | TreeVAE (unsupervised) | HDTree (unsupervised) |
|---|---|---|---|---|---|---|
| LineageVAE Dataset | Day 2 | 22.1% | 2.2% | 16.1% | 12.1% | **23.2%** (↑ +1.1%) |
| | Day 4 | 21.4% | 37.4% | 28.2% | 30.4% | **38.4%** (↑ +1.0%) |
| | Day 6 | 56.6% | 60.3% | 53.2% | 56.4% | **62.0%** (↑ +1.7%) |
| *C. elegans* Dataset | 100–300 | – | 15.4% | 10.8% | 13.8% | **15.2%** (↑ −0.2%) |
| | 300–500 | – | 41.5% | 40.6% | 32.5% | **45.8%** (↑ +4.3%) |
| | 500–750 | – | 62.3% | 51.8% | 62.8% | **66.3%** (↑ +4.0%) |

verify if the model has captured the underlying biological dynamics. **Analysis:** (1) On MNIST, generating digits 3, 1, and 9 from 6 demonstrates smooth and continuous transitions. (2) On *C. elegans*, the stem-to-somatic generation captures consistent gene expression trends, visualized via pie charts.

**Case Study on *C. elegans* Lineage [better accuracy].** To evaluate the performance of HDTree on real lineage labels, we utilized labeled data provided by (Packer et al., 2019). The *C. elegans* dataset not only labels the type of cell but also the relative time at which the cell is collected, which can be regarded as the gold label for our analysis of the cell's differentiation lineage. HDTree demonstrates superior performance compared to TreeVAE in capturing lineage relationships and generating biologically meaningful results. A detailed introduction is in the caption of Fig. 3. **Analysis:** (1) By analyzing Fig. 3(a) and Fig. 3(d), both TreeVAE and HDTree can distinguish different cell types well because they map cells of the same type to similar locations. (2)

TreeVAE cannot accurately model the differentiation process of cells. This is because the differentiation lineage visualization (Fig. 3(b)) based on the TreeVAE representation does not match the real-time gold label (Fig. 3(c)). In contrast, the differentiation lineage inferred by HDTree better aligns with the time gold label (Fig. 3(f)).

**Comparisons on Computational Cost [better efficiency].** To evaluate the computational efficiency of HDTree, we compared the training time of HDTree with TreeVAE and TreeVAE[A] on four datasets (in Table 5). We observe that traditional methods have an advantage on small datasets but scale poorly to large ones. HDTree demonstrates competitive training efficiency, particularly on larger datasets like Weinreb, where it outperforms TreeVAE significantly. It is important to note that while HDTree is efficient in training and embedding (crucial for lineage analysis tasks), its generative inference is inherently slower than VAE-based methods due to the iterative denoising process of diffusion models. This represents a trade-off where we prioritize

*Table 4.* **Ablation Study on MNIST & ECL Datasets.** We evaluate the impact of key components: Hierarchical Tree Codebook (HTC), Soft Contrastive Learning (SCL), and Hierarchical Quantization Loss (HQL). The best performance is highlighted in **bold**.

| Ablation Setups | MNIST (General Dataset) | | | | ECL (Single-Cell Dataset) | | | |
|---|---|---|---|---|---|---|---|---|
| | DP(↑) | LP(↑) | ACC(↑) | NMI(↑) | DP(↑) | LP(↑) | ACC(↑) | NMI(↑) |
| A1. Full Model (Ours) | **92.7** | **97.1** | **96.6** | **92.4** | **69.0** | **83.2** | **83.2** | **79.0** |
| A2. w/o HTC | 87.4 | 85.3 | 84.1 | 75.2 | 58.7 | 71.4 | 70.8 | 66.5 |
| A3. w/o SCL ($\mathcal{L}_{SCL}$) | 78.9 | 82.1 | 81.5 | 73.1 | 55.6 | 68.9 | 68.3 | 63.1 |
| A4. w/o HQL ($\mathcal{L}_{HQL}$) | 84.1 | 89.3 | 86.8 | 81.7 | 61.4 | 74.2 | 73.7 | 69.4 |

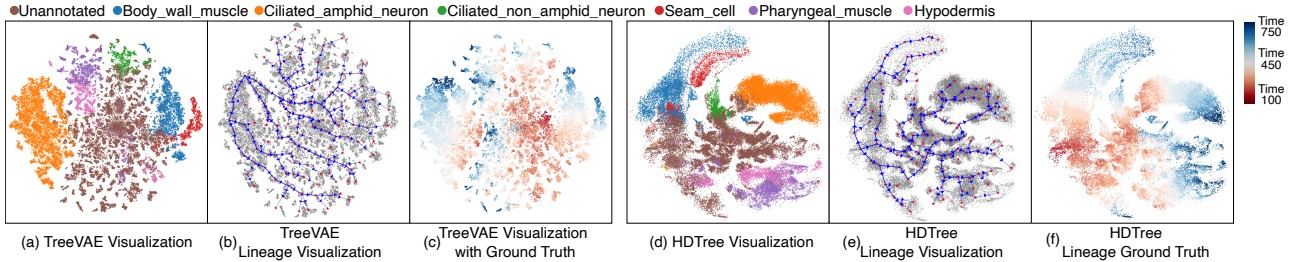

*Figure 3.* **Comparison of TreeVAE and HDTree methods for visualization and lineage inference.** (a) and (d) are the latent space visualization of TreeVAE and HDTree, color shows the cell type information. (b) and (e) are the lineage structure inferred by TreeVAE and HDTree, overlaid on the data distribution. The blue arrows indicate the inferred lineage relationships. (c) and (f) are the ground truth lineage visualization for comparison, the color shows the real-time information (from blue to red). HDTree captures more accurate lineage relationships and generates more realistic data than TreeVAE.

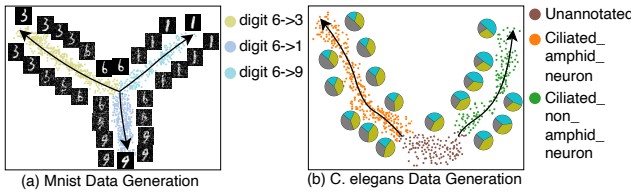

*Figure 4.* **Data generation of HDTree on MNIST and *C. elegans*.** Each scatter is the generated data visualized by tSNE. Color indicates the label. For MNIST, data is generated from digit 6 to 3, 1, and 9. For *C. elegans*, from stem cell to somatic cell (marker genes shown in pie charts: **odr-10**, **osm-6**, **elt-5**.)

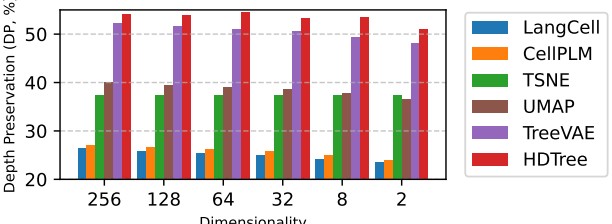

*Figure 5.* **Sensitivity Analysis of Dimensionality on Dendrogram Purity (DP) Across Methods.** The performance of various methods on the ECL dataset with different latent dimensionalities.

*Table 5.* Training time comparison on general and single-cell datasets. **Bold** denotes the best result. (mm:ss)

| | tSNE+Agg | UMAP+Agg | TreeVAE | HDTree |
|---|---|---|---|---|
| MNIST | 14:14 | **2:09** | 192:09 | 42:23 |
| F-MNIST | 15:15 | **2:22** | 206:13 | 45:02 |
| LHCO | 28:28 | **13:51** | 246:20 | 53:23 |
| Weinreb | 97:59 | 340:18 | 361:12 | **53:47** |

higher-fidelity lineage modeling and more stable hierarchical representations over raw sampling speed. Future work could explore distillation techniques to accelerate inference.

**Ablation Study [better accuracy].** To evaluate the contributions of HDTree's components, we conducted ablation experiments on MNIST and ECL datasets. The setups included **(A1)** Full Model (HDTree), **(A2)** without the HTC

and directly use vanilla codebook, **(A3)** without the SCL ($\mathcal{L}_{SCL}$) and directly use the contrastive learning loss, and **(A4)** without the HQL loss ($\mathcal{L}_{HQL}$) and directly use the VQ Loss. Performance is measured using tree structure (*DP, LP*) and clustering metrics (*ACC, NMI*). The results are shown in Table 4. **Analysis:** The full model consistently achieved the best performance. Removing the HTC (A2) caused the most significant performance drop across both datasets, highlighting its role in structural and clustering performance. The SCL (A3) is essential for maintaining tree depth and clustering interpretability. All components contribute to HDTree's success, with the HTC and contrastive loss being the most critical for optimal performance.

**Parameter Sensitivity Analysis [better stability].** To evaluate the impact of latent dimensionality on model perfor-

mance, we conducted a sensitivity analysis using tree performance (DP) as the primary metric. The baseline and SOTA methods are evaluated on the ECL dataset with varying latent dimensionalities. The results are presented in Fig. 5. **Analysis:** HDTree consistently outperforms competing methods across all latent dimensionalities, achieving optimal performance in the 64-256 range while effectively avoiding over-parameterization and robustly preserving hierarchical structures.

## 5. Conclusion

We introduce HDTree, a unified diffusion-based framework for hierarchical representation and generative lineage modeling. By combining a quantized diffusion process with a hierarchical codebook, HDTree captures tree-structured relationships without relying on branch-specific modules, leading to enhanced stability, inference accuracy, and interpretability. Experimental results demonstrate consistent improvements in clustering accuracy, tree structure fidelity, and lineage alignment. **Limitations.** Although HDTree achieves high-fidelity lineage simulation, its diffusion-based decoder remains computationally expensive during sampling, especially for large-scale datasets. Future work will focus on accelerating generation via fast-sampling strategies and efficient latent diffusion schemes.

## Acknowledgements

This work was supported in part by National Science and Technology Major Project of China (No.2021YFA1301603), National Science and Technology Major Project (No. 2022ZD0115101), National Natural Science Foundation of China Project (No. U21A20427), Project (No. WU2022A009) from the Center of Synthetic Biology and Integrated Bioengineering of Westlake University, Zhejiang Province Selected Funding for Postdoctoral Research Projects (No. ZJ2025113), and InnoHK Program. We thank the AI Station of Westlake University for the support of GPUs. We also thank the Funding from the ROOTCLOUD TECHNOLOGY CO.,LTD.

## Impact Statement

This work contributes to the fields of hierarchical representation learning and AI for Science. The primary positive impact of HDTree lies in its ability to reconstruct complex cellular lineages, offering researchers a powerful tool to understand biological development and heterogeneity. This facilitates data-driven discoveries in genomics that may impact healthcare research. Potential negative impacts are primarily related to the reliability of generative models; generated scientific hypotheses (e.g., inferred lineage paths) should be treated as predictions requiring experimental vali-

dation to avoid misinterpretation of biological phenomena. We do not foresee immediate negative societal consequences or misuse cases beyond standard data privacy concerns inherent to biomedical datasets.

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

# A. Appendix: Details of Related Work

**Tree-Structured Representation & Generative Models.** Tree-structured representations are essential for modeling hierarchical relationships within data (Zang et al., 2025c). Traditional approaches, such as hierarchical clustering (Müllner, 2011) and distance-based methods (Kaufman & Rousseeuw, 2009; Bouguettaya et al., 2015), use predefined metrics to construct trees and have been foundational in many applications. Recent advancements in deep learning have introduced adaptive techniques for tree construction, such as the Nouveau VAE (NVAE)(Vahdat & Kautz, 2020), which captures hierarchical semantics through recursive structures, and the Tree Variational Autoencoder (TreeVAE)(Manduchi et al., 2023), which encodes hierarchical latent structures in generative models. Furthermore, hyperbolic geometry has emerged as a powerful framework for representing hierarchical relationships, with methods like Poincaré Embeddings(Nickel & Kiela, 2017) and Hyperbolic Graph Neural Networks (HGNNs)(Zhou et al., 2023) offering efficient and expressive representations of such structures. TreeVI extends variational inference by using a tree structure to efficiently capture correlations among latent variables in the posterior, enabling scalable reparameterization and training while improving performance in tasks like constrained clustering, user matching, and link prediction. Tree-based generative models offer powerful solutions for modeling hierarchical relationships and multi-modal data distributions. GAN-Tree(Kundu et al., 2019) introduces a hierarchical divisive strategy with a mode-splitting algorithm for unsupervised clustering, effectively addressing mode-collapse and discontinuities in data, while enabling incremental updates by modifying only specific tree branches.

**Deep Learning Based Cell Lineage Analysis.** Cell lineage analysis is a vital task in single-cell genomics, aiming to reconstruct developmental trajectories of cells. Traditional methods like *Monocle*(Trapnell et al., 2014) and *Slingshot*(Street et al., 2018) infer pseudotime trajectories but are limited by reliance on predefined metrics and inability to model unobserved progenitor states. Recent advances, such as *LineageVAE*(Majima et al., 2024) and *Waddington-OT*, address these limitations using probabilistic models and optimal transport, yet often face challenges with the high dimensionality and sparsity of single-cell data. Generative models like *Diffusion Pseudotime Models*(Haghverdi et al., 2016) and *TrajectoryNet*(Tong et al., 2020) enhance scalability and interpretability but typically lack mechanisms to model hierarchical relationships in differentiation. Our proposed HDTree integrates hierarchical tree structures into the generative process, enabling accurate reconstruction of both observed and unobserved cell states while improving the interpretability of cell lineage trajectories.

# B. Appendix: Details of Dataset

## B.1. Datasets

In this section, we provide an overview of the datasets used in our evaluation. We consider a diverse set of datasets, including image, text, and single-cell data, to assess the performance of hierarchical clustering methods across different domains. The datasets are selected based on their popularity, complexity, and relevance to real-world applications. We provide a brief description of each dataset, along with key statistics and preprocessing steps.

**MNIST:** A widely-used dataset consisting of 70,000 grayscale images of handwritten digits, each of size $28 \times 28$. Each image is flattened into a 784-dimensional vector. This dataset is primarily used for benchmarking tree structure modeling and hierarchical clustering methods. More details can be found at [2].

**Fashion-MNIST:** A dataset of 70,000 grayscale images representing 10 categories of clothing items, each with a resolution of $28 \times 28$. The images are flattened into 784-dimensional vectors for analysis. This dataset is used to evaluate the robustness of methods on visual data with more complex patterns than MNIST. Dataset details are available at [3].

**20News-Groups:** A text dataset with 18,846 newsgroup posts across 20 categories. The features are represented as TF-IDF vectors with dimensionality reduced to 2,000 features for computational feasibility. This dataset is employed to test methods on high-dimensional text data and hierarchical categorization tasks. More details can be found at [4].

**CIFAR-10:** A dataset of 60,000 color images spanning 10 classes, with each image sized at $32 \times 32$. The pixel intensity values are flattened into a 3,072-dimensional vector. This dataset is utilized for evaluating hierarchical modeling methods on high-dimensional image data. Details are available at [5].

---

[2]http://yann.lecun.com/exdb/mnist/

[3]https://github.com/zalandoresearch/fashion-mnist

[4]http://qwone.com/ jason/20Newsgroups/

[5]https://www.cs.toronto.edu/ kriz/cifar.html

*Table 6.* **Summary of Dataset Statistics.**

| Dataset | Type | Class | Samples | Features | Source URL or Reference |
|---|---|---|---|---|---|
| MNIST | Image | 10 | 70,000 | 784 | `http://yann.lecun.com/exdb/mnist/` |
| Fashion-MNIST | Image | 10 | 70,000 | 784 | `https://github.com/zalandoresearch/fashion-mnist` |
| 20News-Groups | Text | 20 | 18,846 | 2,000 | `http://qwone.com/~jason/20Newsgroups/` |
| CIFAR-10 | Image | 10 | 60,000 | 3,072 | `https://www.cs.toronto.edu/~kriz/cifar.html` |
| Limb | Single-cell | 10 | 66,633 | 500 | `https://limb-dev.cellgeni.sanger.ac.uk/` |
| LHCO | Single-cell | 7 | 10,628 | 500 | `https://www.ebi.ac.uk/biostudies/arrayexpress/studies/E-MTAB-10973` |
| Weinreb | Single-cell | 11 | 130,887 | 500 | `https://www.ncbi.nlm.nih.gov/geo/query/acc.cgi?acc=GSM4185642` |
| ECL | Single-cell | 10 | 838,000 | 500 | `https://cellxgene.cziscience.com/collections/45d5d2c3-bc28-4814-aed6-0bb6f0e11c82` |

**Limb:** A single-cell RNA-seq dataset collected from limb bud development experiments. After selecting the 10 most frequent cell types and the top 500 highly variable genes, the processed dataset used in our experiments contains 66,633 cells with 500 features per cell. This dataset is used to study developmental trajectories in biological processes. Dataset details are available in (Zhang et al., 2024).

**LHCO:** Single-cell transcriptomics data derived from lung and heart cell ontogeny studies. The features represent gene expression profiles across 15,000 cells with dimensionality reduced to 10,000 genes. This dataset is used to explore differentiation patterns in complex organ systems. Dataset details can be found in (He et al., 2022).

**Weinreb:** A lineage-specific dataset focused on Darwinian lineage inference from single-cell data. It contains 8,000 cells with high-dimensional transcriptomic data preprocessed to 12,000 genes. This dataset is employed for benchmarking methods for lineage tracing tasks. Dataset details are provided in (Weinreb et al., 2020).

**ECL:** Single-cell data from embryonic cell lineages, designed for studying early developmental processes. It comprises 12,000 cells with 15,000 genes per cell after quality filtering. This dataset is used to test hierarchical modeling on datasets with complex lineage structures. Details are available in (Qiu et al., 2024).

The key characteristics of the datasets are summarized in Table 6.

### B.2. Dataset Organization

To ensure reproducibility, the datasets were organized according to the following directory structure.

For image datasets:

**Image Datasets Directory Structure**

```
datasets/
|-- MNIST/
|    |-- train-images-idx3-ubyte
|    |-- train-labels-idx1-ubyte
|    |-- t10k-images-idx3-ubyte
|    \-- t10k-labels-idx1-ubyte
|-- FashionMNIST/
|    |-- train-images-idx3-ubyte
|    |-- train-labels-idx1-ubyte
|    |-- t10k-images-idx3-ubyte
|    \-- t10k-labels-idx1-ubyte
|-- 20news/
|    |-- alt.atheism/
|    |    |-- 12345.txt
|    |    |-- 67890.txt
|    |    \-- ...
|    |-- comp.graphics/
|    |    |-- 12346.txt
|    |    |-- 67891.txt
|    |    \-- ...
|    \-- ...
\-- cifar-10-batches-py/
     |-- data_batch_1
     |-- data_batch_2
     |-- data_batch_3
     |-- data_batch_4
     |-- data_batch_5
     |-- test_batch
     \-- batches.meta
```

For biological datasets:

**Biology Datasets Directory Structure**

```
datasets_bio/
|-- original/
| |-- EpitheliaCell.h5ad
| |-- LimbFilter.h5ad
| |-- He_2022_NatureMethods_Day15.h5ad
| |-- Weinreb_inVitro_clone_matrix.mtx
| |-- Weinreb_inVitro_gene_names.txt
| |-- Weinreb_inVitro_metadata.txt
| - Weinreb_inVitro_normed_counts.mtx
- processed/ (exists once the process is run)
|-- EpitheliaCell_data_n.npy
|-- EpitheliaCell_label.npy
|-- LimbFilter_data_n.npy
|-- LimbFilter_label.npy
|-- LHCO.h5ad
- Weinreb.h5ad
```

## B.3. Preprocessing

Before training, datasets need to be preprocessed. Preprocessing steps differ depending on the type of dataset. Below are the detailed guidelines:

### B.3.1. IMAGE DATA

For image datasets (e.g., MNIST, FMINST), preprocessing is straightforward and can leverage the code from TreeVAE. The steps include: Initially, downloading and organizing the data ensures that all necessary dataset files, including images and corresponding labels, are retrieved from their sources and systematically placed into the appropriate directories within the project structure. This step is essential for maintaining data integrity and facilitating efficient access during subsequent processing stages. Next, converting raw data into NumPy arrays involves using provided scripts to load the raw image and label data. These scripts parse the binary or structured data formats and convert them into NumPy arrays, which are optimized for numerical computations in Python. This conversion facilitates further data manipulation and model training processes by providing a standardized format for handling large-scale datasets. Finally, normalization and formatting of the pixel values are performed. This typically includes scaling the pixel intensities to a range of [0, 1] to standardize the input features across different images. Additionally, any necessary adjustments are made to ensure the data format aligns with the requirements of the HDTree model, such as ensuring correct dimensionality and data type consistency. Proper normalization enhances model convergence and stability during training.

### B.3.2. BIOLOGICAL DATA

For biological datasets (LHCO, Limb, Weinreb, ECL), preprocessing is more complex and tailored to each dataset. Below are the detailed preprocessing steps for each:

**LHCO:** Initially, data cleaning is performed to enhance the quality of the dataset. This includes the removal of duplicate entries and invalid samples that could introduce bias or noise into subsequent analyses. Additionally, strategies for handling missing values are implemented, which may involve imputation techniques or filtering out rows with an excessive amount of missing data. Following data cleaning, feature extraction is conducted to identify and extract relevant features from the raw data. For LHCO datasets, these features often include particle kinematics or event-level characteristics that are critical for analysis. Once extracted, feature scaling is applied to normalize or standardize the data, ensuring consistency across different scales and facilitating more efficient model training. Finally, the dataset is split into training, validation, and test sets according to predefined configurations.

This is the key code of our preprocessed methods:

*Listing 1.* Preprocess of Lhco

```
adata = sc.read(f"{input_path}/He_2022_NatureMethods_Day15.h5ad")
sc.pp.highly_variable_genes(adata, n_top_genes=500)
adata = adata[:, adata.var['highly_variable']]
data = adata.X
data = adata.X.toarray()
data = np.array(data).astype(np.float32)
mean = data.mean(axis=0)
std = data.std(axis=0)
data = (data - mean) / std
```

**Limb:** It begins with data loading, where the dataset is read from the provided files—typically in formats such as CSV or HDF5—into a structured representation suitable for further processing. Next, the data undergoes filtering and cleaning to enhance its quality and reliability. This includes the removal of noise or artifacts that may have been introduced during data acquisition, as well as deduplication to eliminate redundant entries. Missing values are addressed through appropriate strategies, such as imputation or selective removal of incomplete records. Following this, feature engineering is performed to transform raw biological signals into more interpretable and informative features. For instance, limb motion patterns or other domain-specific characteristics may be extracted to better capture the underlying structure of the data. These features are then normalized to ensure uniformity in scale, which is essential for many machine learning algorithms. Finally, the dataset is stratified and split into training, validation, and test subsets. This is the key code of our preprocessed methods:

*Listing 2.* Preprocess of Limb

```
adata = sc.read(f"{input_path}/LimbFilter.h5ad")
data_all = adata.X.toarray().astype(np.float32)
label_celltype = adata.obs['celltype'].to_list()
vars = np.var(data_all, axis=0) # HVG
mask_gene = np.argsort(vars)[-500:]
data_hvg = data_all[:, mask_gene]
label_count = {}
for i in list(set(label_celltype)):
    label_count[i] = label_celltype.count(i)
label_count = sorted(label_count.items(), key=lambda x: x[1], reverse=True)
label_count = label_count[:10]
mask_top10 = np.zeros(len(label_celltype)).astype(np.bool_)
for str_label in label_count:
    mask_top10[str_label[0] == np.array(label_celltype)] = 1
data_n = np.array(data_hvg).astype(np.float32)[mask_top10]
mean = data_n.mean(axis=0)
std = data_n.std(axis=0)
data = (data_n - mean) / std
```

**Weinreb:** Initially, data transformation is performed to normalize the raw gene expression counts. This typically includes log-transformation or conversion into normalized values such as Counts Per Million (CPM), Transcripts Per Million (TPM), or Fragments Per Kilobase of transcript per Million mapped reads (FPKM). Low-expression genes, as well as cells with insufficient sequencing depth or missing data, are filtered out to improve signal-to-noise ratio and computational efficiency. Following normalization, dimensionality reduction techniques—such. These methods reduce the feature space while preserving the major sources of variation in the data, which can improve model performance and reduce computational burden. When the dataset originates from multiple experimental batches or sources, batch effect correction is employed to mitigate technical variability that could confound biological signal detection. Various statistical or machine learning-based approaches may be used depending on the nature of the data and experimental design. Finally, the dataset is stratified and partitioned into training, validation, and test sets.

This is the key code of our preprocessed methods:

*Listing 3.* Preprocess of Weinreb

```
matrix_file = f"{input_path}Weinreb_inVitro_normed_counts.mtx"
genes_file = f"{input_path}Weinreb_inVitro_gene_names.txt"
metadata_file = f"{input_path}Weinreb_inVitro_metadata.txt"
mtx = mmread(matrix_file).tocsr()
genes = pd.read_csv(genes_file, header=None, names=['genes'])
adata = sc.AnnData(mtx, var=genes)
metadata = pd.read_csv(metadata_file, sep='\t')
adata.obs = metadata.set_index(adata.obs.index)
adata.write(f'{output_path}Weinreb.h5ad')
sc.pp.log1p(adata)
adata.obs['celltype']=adata.obs['Cell type annotation']
adata = adata[~adata.obs['celltype'].isna()]
sc.pp.highly_variable_genes(adata, n_top_genes=500)
adata = adata[:, adata.var['highly_variable']]
data = adata.X.toarray()
data = np.array(data).astype(np.float32)
mean = data.mean(axis=0)
std = data.std(axis=0)
data = (data - mean) / std
```

**ECL:** Initially, raw data files are parsed and converted into structured tabular formats that facilitate further computational

processing. This is followed by a preprocessing stage that includes normalization—commonly achieved through z-score transformation or Min-Max scaling—and the handling of missing values, which may involve either imputation techniques or the removal of incomplete samples. Subsequently, feature selection is performed with an emphasis on retaining biologically meaningful attributes, often guided by domain-specific knowledge such as known molecular signatures. Finally, the dataset is partitioned into training, validation, and test subsets, ensuring that class distributions are preserved across splits to support unbiased model evaluation and generalization.

This is the key code of our preprocessed methods:

*Listing 4.* Preprocess of ECL

```
adata = sc.read(f"{input_path}/EpitheliaCell.h5ad")
adata.obs['celltype']=adata.obs['cell_type']
label_celltype = adata.obs['celltype'].to_list()
adata_sub = adata.copy()
sc.pp.subsample(adata_sub, fraction=0.1)
data_all = adata_sub.X.toarray().astype(np.float32)
vars = np.var(data_all, axis=0)
mask_gene = np.argsort(vars)[-500:]
adata = adata[:, mask_gene]
data = adata.X.toarray().astype(np.float32)
label_count = {}
for i in list(set(label_celltype)):
    label_count[i] = label_celltype.count(i)
label_count = sorted(label_count.items(), key=lambda x: x[1], reverse=True)
label_count = label_count[:10]
mask_top10 = np.zeros(len(label_celltype)).astype(np.bool_)
for str_label in label_count:
    mask_top10[str_label[0] == np.array(label_celltype)] = 1
data_n = np.array(data).astype(np.float32)[mask_top10]
label_train_str = np.array(list(np.squeeze(label_celltype)))[mask_top10]
# downsample the 10k data for every cell type
mask = np.zeros(len(label_train_str)).astype(np.bool_)
for i in range(10):
    # random select 10k data for each cell type
    random_index = np.random.choice(
        np.where(label_train_str == label_count[i][0])[0],
        10000, replace=False)
    mask[random_index] = 1
data_n = data_n[mask]
mean = data_n.mean(axis=0)
std = data_n.std(axis=0)
data= (data_n - mean) / std
```

## C. Appendix: Details of Baseline Methods

Baseline methods include traditional approaches, such as Agglomerative Clustering (Agg) (Müllner, 2011), t-SNE (Linderman & Steinerberger, 2019), and UMAP (Dalmia & Sia, 2021), as well as state-of-the-art (SOTA) deep learning methods, including VAE (Doersch, 2016; Lim et al., 2020), LadderVAE (Sønderby et al., 2016), DeepECT (Mautz et al., 2020), and TreeVAE (Manduchi et al., 2023). Additionally, specialized models (Geneformer (Theodoris et al., 2023), LangCell (Zhao et al., 2024), CellPLM (Wen et al., 2024)) tailored for the single-cell domain are incorporated to ensure a thorough evaluation across diverse tasks.

### C.1. Implementation Details of Baselines

We utilized the official implementations for the baseline methods to ensure a fair comparison.

- **TreeVAE**: We utilized the official implementation[6] and followed the standard configuration described in Manduchi et al. (2023).

- **CellPLM**: The official implementation[7] was used. We followed the official tutorial for cell embedding extraction.

- **LangCell**: We used the code provided in the official repository[8] and followed the zero-shot annotation workflow.

- **Geneformer**: We utilized the pre-trained models and code from the official repository[9], following the examples for extracting cell embeddings.

## D. Appendix: Details of Testing Protocol

To ensure a comprehensive and reproducible evaluation, we report clustering quality, hierarchical structure quality, and generative/reconstruction quality. Unless noted, metrics are computed on the test split and averaged over multiple runs (with fixed random seeds).

**Clustering Accuracy (ACC).**   ACC is computed via an optimal one-to-one relabeling using the Hungarian algorithm. Let $y_i$ be the ground-truth label and $\hat{y}_i$ the predicted cluster.

$$\text{ACC} = \frac{1}{n} \max_{\pi \in \mathcal{S}} \sum_{i=1}^{n} \mathbf{1}\{y_i = \pi(\hat{y}_i)\},$$

where $\mathcal{S}$ is the set of all label permutations. We use the standard Hungarian implementation to obtain $\pi$.

**Normalized Mutual Information (NMI).**   Given predicted clustering $C$ and ground-truth clustering $G$,

$$\text{NMI}(C, G) = \frac{2\, I(C; G)}{H(C) + H(G)},$$

where $I(\cdot\,;\cdot)$ is mutual information and $H(\cdot)$ is entropy. We use the symmetric NMI with natural logarithms. $\text{NMI} \in [0, 1]$ (higher is better).

**Leaf Purity (LP).**   Let the learned tree $T$ have leaf nodes $\{L_1, \ldots, L_k\}$. For leaf $L_i$, define $L_i^y = \{x \in L_i : \text{label}(x) = y\}$. We report the macro-average over non-empty leaves:

$$\text{LP} = \frac{1}{k} \sum_{i=1}^{k} \frac{\max_y |L_i^y|}{|L_i|}.$$

Empty leaves (no assigned samples) are excluded from the average.

**Dendrogram Purity (DP).**   *Throughout the paper, DP denotes **Dendrogram Purity**, consistent with hierarchical clustering literature and prior work.* For each class $c$, consider all pairs $(i, j)$ with $y_i = y_j = c$. Let $\text{LCA}(i, j)$ be the lowest common ancestor cluster of $x_i$ and $x_j$ in the dendrogram, and let $S_{ij}$ be the set of samples contained in that cluster. The pairwise purity is

$$\text{pur}(i, j) = \frac{|\{p \in S_{ij} : y_p = c\}|}{|S_{ij}|}.$$

DP is the average of $\text{pur}(i, j)$ over all intra-class pairs across all classes. This metric increases when intra-class pairs meet early in the tree (at purer LCA nodes).

---

[6] https://github.com/lauramanduchi/treevae
[7] https://github.com/OmicsML/CellPLM
[8] https://github.com/PharMolix/LangCell
[9] https://github.com/jkobject/geneformer

**Reconstruction Loss (RL).**    Given inputs $X = \{x_i\}$ and reconstructions $\hat{X} = \{\hat{x}_i\}$, we compute MSE:

$$\text{RL} = \frac{1}{n} \sum_{i=1}^{n} \|x_i - \hat{x}_i\|_2^2.$$

To align the "higher-is-better" convention across metrics, we report $-\text{RL}$ in tables (i.e., larger is better). Reconstructions for diffusion models are obtained by conditioning on the learned latent path and running the standard deterministic denoising trajectory at evaluation time.

**Log-Likelihood (LL) for Diffusion Models.**    Exact log-likelihood is intractable for DDPMs; we report the negative ELBO (variational lower bound) following standard practice. Concretely, we sum the per-timestep KL (or reweighted MSE) terms under the chosen $\{\beta_t\}$ schedule and include the analytic prior and decoder terms as in Ho et al. (2020). We report the per-sample LL (higher is better). Implementation matches our training loss with the appropriate constants added back.

**Fréchet Inception Distance (FID).**    *Images:* we compute FID in the 2048-D Inception-V3 pool3 feature space, matching the number of generated and real samples and using the same preprocessing. *Single-cell (scRNA-seq):* we compute FID in a biologically meaningful feature space: (i) select HVGs (e.g., top-1,000 by variance) on the training set; (ii) normalize real and generated matrices identically; (iii) optionally correct batch effects (e.g., Harmony/Scanorama) *before* feature extraction; (iv) run PCA to retain $> 90\%$ variance (typically $\sim$50 PCs); (v) estimate Gaussians in the PC space and compute FID via covariance square roots (with a small diagonal regularizer if needed). We fix random seeds and average FID over multiple generations.

**Ratio of Observed Time Points (ROP) for Lineage Consistency.**    For time-resolved single-cell datasets, we quantify local temporal coherence by measuring, for each cell, the fraction of its $k$-nearest neighbors (in the learned representation) whose time stamps are consistent with its developmental order; we then average over all cells and report by time window as in the main paper. Ablations show ROP strongly correlates (negatively) with tree-edit distance, supporting its biological relevance.

**Implementation Notes (All Metrics).**    (i) ACC relabeling uses the Hungarian algorithm; ties are broken deterministically. (ii) Empty leaves are excluded from LP; singleton leaves contribute 1.0. (iii) All metrics are averaged across $r$ runs (defaults given in code) with fixed seeds and identical preprocessing. (iv) For diffusion metrics (RL/LL/FID), generation uses the same $\{\beta_t\}$ schedule and evaluation pipeline across methods.

# E. Appendix: Details of Implementation

For all experiments, the data is split into training, validation, and testing sets with an 8:1:1 ratio, ensuring unbiased evaluation. In testing, if the number of points in the dataset is greater than 10,000, we randomly sample 10,000 points from testing dataset. Details on downsampling and its rationale are provided in the Appendix. We implemented HDTree using PyTorch and trained the model on a single NVIDIA A100 GPU. The model is trained using the AdamW optimizer with a cosine learning-rate schedule. The number of diffusion steps $T$ is set to 1000, and the tree depth $L$ is set to 10. We use dataset-specific batch sizes and learning rates selected on validation performance; for the public MNIST and Limb configurations, the batch sizes are 5000 and 1000, respectively, and the learning rate is 0.005. The loss weights and routing/exaggeration parameters are specified in the released configuration files. The encoder and diffusion model are implemented using multilayer perceptrons (MLPs).

# F. Appendix: Details of Downsampling in Testing

As shown in Table 7, the computational time required for encoding, dimensionality reduction (LowDim), and clustering (including evaluation metrics computation) significantly increases with larger data sizes. For instance, when the input size increases to N=100,000, the encoding time for UMAP rises from 820s to 3153s, and the clustering and evaluation time for GeneFormer increases from 25s to 3285s. These trends are consistent across all tested methods.

Such exponential growth in computational overhead makes the evaluation process infeasible for large-scale datasets. To address this challenge, we adopt a conditional downsampling strategy: we only downsample the test set to 10,000 points if the number of test samples exceeds 10,000. Given our 8:1:1 data split, this threshold is only triggered when the total dataset size

*Table 7.* **Time Efficiency Comparison Across Methods and Data Sizes (seconds):** We conducted an empirical evaluation by selecting varying numbers of highly expressed channels to assess the computational time cost of our model under different data sizes. For this experiment, we set the output dimensionality of the Embedding method to 512 dimensions and employed UMAP for dimensionality reduction to facilitate visualization. The resulting features were then subjected to clustering analysis. As the dataset size increased from 10,000 to 100,000 samples, we observed a dramatic increase in the computational time required for both the dimensionality reduction and clustering processes. Note: The reported time includes both clustering and the computation of evaluation metrics. The extended duration is primarily due to the Hungarian algorithm required for calculating Clustering Accuracy (ACC) on large-scale datasets. Consequently, we opted to downsample the data to 10,000 samples for further processing.

| Method | PCA | TSNE | UMAP | CellPLM | LangCell | GeneFormer |
|---|---|---|---|---|---|---|
| **Cin=100, C=512, N=10000** | | | | | | |
| Embedding | None | None | 820s | 6s | 13s | 27s |
| DR | None | None | 120s | 129s | 133s | 150s |
| Clustering & Evaluation | None | None | 41s | 37s | 30s | 25s |
| **Cin=1000, C=512, N=10000** | | | | | | |
| Embedding | 140s | None | 800s | 7s | 13s | 25s |
| Dimensional Reduction | 140s | None | 120s | 130s | 130s | 105s |
| Clustering & Evaluation | 45s | None | 40s | 37s | 34s | 30s |
| **Cin=1000, C=512, N=100000** | | | | | | |
| Embedding | 224s | None | 3153s | 13s | 11s | 220s |
| Dimensional Reduction | 413s | None | 825s | 416s | 430s | 420s |
| Clustering & Evaluation | 9637s | None | 7083s | 5037s | 2032s | 3285s |
| **Using K-means to 5000 clusters** | | | | | | |
| Embedding | 221s | None | 1834s | 14s | 123s | 110s |
| Dimensional Reduction | 458s | None | 411s | 350s | 427s | 410s |
| Clustering & Evaluation | 2925s | None | 20429s | 5500s | 3432s | 2200s |

exceeds 100,000 samples. Consequently, for the majority of our benchmarks (MNIST, Fashion-MNIST, 20News-Groups, CIFAR-10, Limb, LHCO), we evaluate on the full test set without any downsampling. Only for the largest datasets (Weinreb and ECL) is this cap applied. This approach is crucial not only for Clustering Accuracy (ACC), which requires the Hungarian algorithm with cubic complexity $O(N^3)$, but also for tree structure metrics. Specifically, the Dendrogram Purity (DP) metric involves calculating the purity of the Lowest Common Ancestor (LCA) for all pairs of samples within the same class, leading to a complexity that grows quadratically with the number of samples per class ($O(N_c^2)$). For datasets like ECL (838k samples) and Weinreb (130k samples), performing pairwise traversals on the full test set is computationally prohibitive.

To ensure that this downsampling does not mask rare populations or introduce bias, we conducted a detailed sensitivity analysis, provided in Appendix **??** (Table **??**). The results show that as the sample size increases from 10k to 300k, the model's ACC and DP metrics remain highly stable, with DP fluctuating only within approximately $\pm 1\%$. This strongly demonstrates that 10,000 sampled points are sufficient to capture the core distribution structure and characteristics of rare populations, ensuring that our evaluation results are unbiased and robust.

