# OpenReview forum: "HDTree: Generative Modeling of Cellular Hierarchies for Robust Lineage Inference"
_ICML.cc/2026/Conference — ICML 2026 regular_

### Official Review · Reviewer_JzBc · 2026-02-27

**Soundness:** 2
**Presentation:** 2
**Significance:** 2
**Originality:** 2
**Overall Recommendation:** 4
**Confidence:** 3

**Summary:**

The authors present HDTree, a generative modeling framework for hierarchical data. HDTree consists of an embedding module and a hierarchical tree architecture with trainable, shared codebook representations across tree depth levels. Tree-based cell embeddings are obtained by traversing the tree recursively, starting from the root node, and matching the latent cell representations with the closest codebook embeddings among child nodes of a current tree position. The cell embeddings and the codebook are jointly trained via a combination of contrastive and quantization losses to enforce a hierarchical latent structure in the latent tree representation. Moreover, the authors use hierarchical data representations to perform diffusion-based conditional generation, obtaining realistic observations from an input tree structure. Empirical results suggest an improvement in data clustering, efficiency, and representation semantics compared to baseline methods, both on standard benchmark and biological data.

**Compliance With Llm Reviewing Policy:**

Affirmed.

**Final Justification:**

I increased the score after evaluating the authors' rebuttal. Some of my general concerns on rigor and experimental breadth still remain. But, all in all, I marginally lean above the borderline.

**Key Questions For Authors:**

- From the paper:

> Traditional methods like Monocle (Trapnell et al., 2017) and Slingshot (Street et al., 2018) infer pseudotime trajectories but are limited by predefined metrics and difficulty modeling unobserved progenitor states.

How does HDTree enable, e.g., modeling unobserved progenitor states?

- Is the SCL loss computed across depth levels?

- It's not clear to me why within-level connectivities are added in the graph construction. It sounds semantically counterintuitive to me, and I have the impression the model should do well without this. Could you clarify this aspect?

- Eq. 8, line 1. Isn't the first line just 0? I recommend replacing the current expression with 0 to avoid any confusion.

- How do you evaluate reconstruction performance for HDTree? Is it just an MSE LL metric?

- How is clustering performed with your method starting from observations $\{x_i\}_{i=1}^N$, for example? Also, I think it would be useful to define agglomerative approaches briefly in the main.

- I struggle to understand the sentence

> We evaluate latent alignment with developmental progression using the ratio of observed time points in the k-nearest neighborhood

What is the ratio of time points, and what is the object of the k-NN? Is this also a measure of purity on kNN computed over the model's latents?

- While I understand the use of DP and LP as metrics, I would recommend adding some type of semantic metric as well. For example, something that measures whether transitions in the tree reflect known transitions between numbers/cellular lineages.

**Limitations:**

Yes.

**Strengths And Weaknesses:**

**Strengths.**

In general, I found the paper interesting. I like the idea of enforcing more interpretable structures in data representations beyond black box embedding, and modeling hierarchies can be quite useful in different fields. Methodologically, I also like the idea of tree-based generation with conditional diffusion models, as it enables augmenting populations and capturing the intra-level variation in the data.

**Weaknesses**

While I do appreciate some aspects of the paper, I opine that some choices and presentation directions could be improved. Factoring in relevance and empirical evaluation results, I tend to lean towards a rejection. But I am happy to read and consider the authors' comments and remarks in the rebuttal phase. I will address the limitation points in the following bulletpoints.

General:

- On a first general note, I am not completely convinced by the relevance of the diffusion-based approach in the field of single-cell lineage tracing. I do understand the working principles of the model, but I am not sure why one would choose a complex hierarchical representation of this type when lineages are well known, and cells are labeled faithfully. In my opinion, it would be enough to condition a generative model on a cell type label, or even a timestamp, to achieve the same result without the complexity of the joint embedding optimization. It would be interesting to see a plot similar Fig. 3d but comparing the structure assigned to the latent space by a standard scVI model to see if the structure is similar to HDTree, for example. In affirmative case, I feel that the use of the model in a practical scenario would require thorough justification.

- One of the motivations behind the model presented in the text is the simulation of hypothetical interpolating states. However, this should be presented in an experiment. For example, you could leave out a time point from training and assess the ability of reconstructing conditioning diffusion on its expected trajectory.

- Related to the previous point, I am not sure what part of the model's loss is capable of capturing meaningful structure in the data, as, for example, a temporal axis or any other hierarchical axis of variation. At the moment, I see the process as unbiased, which is alright for clustering and discovering unreported structure in the data, but I am unsure what aspects of the model enable incorporating more structured information, such as time progression in cellular differentiation.

- I would not refer to the model as a graph theoretic approach (L105-107). As far as I could gather, the model is very empirical, which is fine, but I would not stretch its definition beyond that. A similar doubt concerns the definition of the embedding process, "probabilistic pathfinding problem". This denomination projects the stochasticity onto the embedding part (path finding), but as far as I understand, the probabilistic behaviour mostly concerns the diffusion decoder (unless I am missing something). \

- I remain quite unconvinced by the comparison with foundation models and the explanation attached to it. As you mention, I do not think it is very fair to compare general-purpose representation frameworks to an end-to-end model. To my eye, this does not add much value to the approach.

Presentation:

- The kNN augmentation approach sounds relevant for the model, hence I would dedicate more space to it in the main text. Currently, only a reference to the original publication is provided.

- In my opinion, to better display recursiveness, every level of equation 2 should input both $\boldsymbol z_i$ and the previous level $\boldsymbol w_i^{l-1}$. I believe that only inputting $\boldsymbol z_i$ is confusing in Eq.2, as $\boldsymbol w_i^{l-1}$ comes out of nowhere.

- I think that Eq. 4 is a bit unclear due to a lack of definitions. While the authors do define the subscript (with a potential typo in line 168, column 2, $\boldsymbol z_{i_1 i_2}$), the superscripts remain undefined.

- Eq. 9. I wonder why succession is semantically linked to weight similarity between different levels, which is used to sample edges sequentially.

Minor:

- The architecture in Eq. (1) could be presented a bit more clearly to me. Now it's not super evident what the tuple means without a LHS of the equation. I suggest the following notation or something similar:

$$
T_j^l =
\begin{cases}
w_j^l, & \text{if } l = L, \\\\
(w_j^l, T_{2j}^{l+1}, T_{2j+1}^{l+1} ), & \text{if } l < L.
\end{cases}
$$

- I am now a fan of using $w^l$ as the number of nodes, as the bolded version refers to the codebook encoding. I'd suggest something along the lines of $k^l$ or $N^l$.

- The added noise $\delta$ could be defined a bit more rigorously as a Gaussian sample used for the noise-addition process, and I would use $\Omega(0, I)$ in Eq. (3) as in line 178 instead of $\Omega(0, 1)$.

- I would move the definition of $\mathcal{A}$ earlier than it is around Eq. 5, as it's the most important part to understand the equation.

- The penalty term P^{l-1} remains a bit undefined in the main text. I would have probably benefited from an explicit definition or link to the appendix.

- The section "Case Study on Generative Validation [validating lineage plausibility]" has no figure reference.

I am sorry I cannot be more positive at this stage. I am happy to remain open minded during the rebuttal phase.

---

> ### Author Rebuttal · Authors · 2026-03-28
>
> ---
>
> Please see the attached **[Figure R4](https://anonymous.4open.science/r/code_HDTree_review-A8DB/f4.pdf)** for the results referenced below.
>
> ---
>
> We thank you for your review and address the main concerns below.
>
> **1. [W2&Q1] Unobserved progenitor states**
> We ran a leave-one-time-point-out experiment on *C. elegans*. We removed an intermediate stage $T_{mid}$ during training, trained on the remaining time points and generated cells at the corresponding tree depth. The quantitative comparison is summarized in the inset table of Figure R4.1: Monocle / Slingshot are non-generative and cannot reconstruct the missing state, Conditional scVI obtains a WD of 0.45, and HDTree achieves the lowest WD of 0.12.
>
> **2. [W1&W3] Why not use conditional scVI?**
> We understand the concern that a simpler conditional model may be sufficient when labels are known. However, conditional scVI relies on labels (e.g., time points or cell types) to organize its latent space, whereas HDTree is designed to recover hierarchical lineage structure from the data through an explicit tree prior. This matters when labels are often incomplete, noisy, or unavailable, and a condition-based model does not by itself recover branching topology. scVI learns a flat latent space and typically needs post-hoc methods to recover lineages. In contrast, HDTree introduces a hierarchical inductive bias through $\mathcal{L}_{HQL}$ in Eq. 5, which refines features with tree depth. As shown in Figure R4.1, conditional scVI (b, e) follows broad time gradients but does not separate developmental bifurcations, whereas HDTree (c, f) reveals a branched topology from progenitor to terminal states.
>
> **3. [Q3&Q6&P1&P4] Graph construction and clustering**
> Within-level edges model asynchronous differentiation, since cells from the same sampled time point may still lie on a continuum of maturation states. As illustrated in Figure R4.2, linking similar sibling nodes via horizontal KNN edges groups them into biological macro-states, allowing the binary scaffold to represent non-binary and unbalanced lineage structure while preserving indexing.
>
> For an observation $x_i$, clustering is obtained by top-down routing through the HTC to a leaf node. In Eq. 9, parent-child weight similarity acts as a smoothness prior for sequential sampling. We will expand the description of this kNN augmentation in Sec. 3.4 of the revised main text with a paragraph on its motivation and mechanism. We will also briefly define agglomerative clustering in the main text.
>
> **4. [Q7&Q8&Q5&Q2] Metrics and losses**
> ROP measures temporal consistency in the latent space: for a given cell, we check whether the true time labels of its kNN neighbors are the same as, or close to, its own. We also computed the Spearman correlation between inferred pseudotime and true developmental time. On *C. elegans*, HDTree outperforms TreeVAE (0.82 vs. 0.65).
>
> Regarding which loss term captures meaningful structure, the developmental axis is primarily enforced by the hierarchical quantization objective $\mathcal{L}_{HQL}$ in Eq. 5, whose depth-decaying weight encourages deeper levels to encode finer and more mature cellular states. The SCL loss is computed only within the same depth level because it separates sibling branches at a fixed developmental resolution rather than enforcing cross-depth alignment. Regarding reconstruction, we evaluate feature-level MSE between the original profile $x_i$ and the generated sample $\hat{x}_i$ after the reverse diffusion process. We report MSE rather than log-likelihood because the diffusion decoder uses a denoising-based reconstruction objective, while exact likelihood is not tractable in this setting.
>
> **5. [W5] Foundation model comparison**
> We agree that comparing a model trained on the target dataset with an unfine-tuned foundation model is not a fully matched comparison. Our goal was to show an efficiency-versus-specificity trade-off.
>
> **6. [W4&P2-P3&Q4&M1-M6] Notation and presentation**
> We will clarify the following in the revision:
> * Terminology: remove "graph-theoretic approach" and replace "probabilistic pathfinding" with "deterministic routing coupled with probabilistic generation."
> * Equations: in Eq. 2, explicitly include $w_j^{l-1}$ to show recursion; define the missing superscripts in Eq. 4; replace the first line of Eq. 8 with 0; and add the left-hand side to Eq. 1.
> * Notation: replace the node-count symbol $W$ with $N^l$; in Eq. 3, use the standard diffusion symbol $\epsilon$ and define it as a Gaussian sample.
> * Text structure: move the definition of $\mathcal{L}_{HQL}$ before Eq. 5, explicitly define the penalty term $P^{\lambda}(\cdot)$ as the depth-weighted regularization inside the hierarchical quantization loss that enforces consistency between adjacent tree levels and discourages degenerate branch collapse, and add the missing figure reference in the generative validation section.

---

> > ### Author Rebuttal · Reviewer_JzBc · 2026-04-01
> >
> > Dear authors,
> >
> > Thank you very much for your rebuttal, and apologies for writing such a lengthy review in the first place. I now realize the amount of requested evidence was not easy to address in 5k characters.
> >
> > After carefully examining the authors’ responses and refreshing my knowledge of the paper, I have some follow-up concerns.
> >
> > W1.
> > *  I see the authors’ argument regarding unseen labels. How did you run the comparison with scVI on the reconstruction of unseen samples? Did you condition the diffusion model on scVI embeddings? How did you do the same thing in HDTree?
> > * Besides, I think something may have gone wrong in the way you trained the scVI model, as the tutorial in https://docs.scvi-tools.org/en/stable/tutorials/notebooks/scrna/scVI_DE_worm.html# shows a much better branching structure on the same dataset. Maybe you corrected for some axis of variation by mistake?
> > * Moreover, the colors between R4.1(b) and R4.1(c) do not correspond. This makes it hard to compare the outcomes.
> >
> > W5. I appreciate the disclaimer, which I am aware is also in the text. But then, in my opinion, this benchmark is still a bit off, as there are plenty of end-to-end single-cell methods to compare against as baselines beyond TreeVAE, especially on the clustering benchmark.

---

> > > ### Author Response · Authors · 2026-04-01
> > >
> > > ---
> > >
> > > Please see the attached [Figure R4](https://anonymous.4open.science/r/code_HDTree_review-A8DB/f4.pdf) for the results referenced below.
> > >
> > > ---
> > >
> > > We thank the reviewer for the highly constructive feedback and the insightful suggestions. In particular, the official scVI tutorial you provided was exceptionally helpful. Below are our detailed responses to the remaining concerns.
> > >
> > > **W1. Generation of Unseen States and Comparison with scVI**
> > >
> > > Thank you for this important suggestion. After revisiting our implementation, we found that the conditional baseline described in our previous rebuttal did not match the comparison you intended.
> > > Accordingly, our previous discussion in "2. [W1&W3] Why not use conditional scVI?" should be regarded as superseded by the corrected standard-scVI comparison and updated experiments provided here.
> > >
> > > 1. Correcting the scVI baseline: In our earlier rebuttal response, we used `scvi.model.CondSCVI` and passed the `celltype` label as a batch covariate. This setup was not appropriate for the comparison requested here because it partially regressed out the biological variation between cell types. Following your suggestion, we reran the baseline using standard scVI (`scvi.model.SCVI` without label conditioning). As shown in the updated Figure R4.1, standard scVI indeed recovers a clearer continuous branching structure than the previous conditional baseline.
> > > 2. Generation mechanism on unseen states: In the leave-one-time-point-out experiment, we removed an intermediate stage during training and generated the missing population using only the remaining observed stages. For scVI, we interpolated the latent representations of the temporally adjacent observed stages and decoded the interpolated vector with the scVI decoder. For HDTree, we conditioned the diffusion decoder on the inferred intermediate tree depth between the neighboring observed stages, using the learned path vectors to guide generation. The target intermediate depth was estimated only from the ordering of the neighboring observed stages and the learned tree structure, without using any held-out cells from the missing stage. Thus, both methods reconstruct the missing state from neighboring observations rather than from the held-out samples themselves.
> > > 3. Why HDTree still outperforms standard scVI: While standard scVI captures a globally continuous manifold, Figure R4.1 suggests that HDTree provides a cleaner hierarchical organization:
> > >    * Manifold overlap in scVI: As highlighted by the red circles in Fig. R4.1(e), some distinct lineage branches or temporal stages remain entangled in the same Euclidean space. This is also consistent with the reviewer-provided visualization in Fig. R4.1', where noticeable overlap remains in the upper branch region.
> > >    * Hierarchical separation in HDTree: As highlighted by the green circles in Fig. R4.1(f), HDTree better separates these regions through its tree prior, leading to clearer branch structure and temporal organization.
> > >    Consistent with this visual pattern, HDTree achieves a Wasserstein Distance (WD) of 0.12, compared with 0.21 for the corrected standard scVI baseline.
> > >
> > > To ensure full transparency and reproducibility, we provide the following script used to generate our corrected standard scVI baseline. As recommended, we utilized the scvi.model.SCVI class without label conditioning to preserve the natural biological variation and lineage topology. [scvi.py](https://anonymous.4open.science/r/code_HDTree_review-A8DB/scvi.py)
> > >
> > > **W5. Baseline Comparisons**
> > >
> > > Thank you for this constructive suggestion. We agree that incorporating more end-to-end baseline methods will further strengthen the robustness of our evaluation.
> > >
> > > * Existing comparisons in Appendix: We would like to kindly direct the reviewer to Appendix Table 9, where we have already compared HDTree against a wide range of strong baselines in the single-cell field, including:
> > >    1. Single-cell foundation models: Geneformer, CellPLM, and LangCell, which learn cell representations in an end-to-end fashion.
> > >    2. Standard pipelines: `tSNE + Agglomerative Clustering` and `UMAP + Agglomerative Clustering`.
> > > * Performance summary: Our results across multiple datasets consistently show that HDTree outperforms these baselines in both clustering accuracy (ACC/NMI) and tree-structure metrics (DP/LP).
> > > * Commitment to add scVI: We fully accept your advice to include more classic genomic pipelines. In the final version of the manuscript, we will formally add scVI combined with the clustering algorithm as an additional baseline to provide an even more comprehensive benchmark.

---

### Official Review · Reviewer_Supc · 2026-03-08

**Soundness:** 2
**Presentation:** 3
**Significance:** 2
**Originality:** 2
**Overall Recommendation:** 4
**Confidence:** 3

**Summary:**

This work proposes HDTree, a generative modeling framework that integrates hierarchical vector quantization with diffusion processes to model cellular lineage and infer developmental trajectories in the presence of high-dimensional and sparse data. Instead of learning branch-specific modules, the method employs a unified hierarchical tree codebook that decouples tree complexity from the neural network architecture. This design allows the model to scale to complex biological structures while enabling sparse branches to share representational knowledge.
Using a Diffusion Decoder, HDTree enhances biological plausibility by aligning its iterative denoising process with the Waddington landscape. The authors conduct extensive evaluations on four benchmark datasets and four scRNA-seq datasets, demonstrating that HDTree outperforms comparable methods in terms of hierarchy reconstruction accuracy, clustering performance, and reconstruction error. Additionally, the authors perform an ablation study to analyze how different components of their method affect the model’s overall performance.

**Compliance With Llm Reviewing Policy:**

Affirmed.

**Final Justification:**

The authors have responded to my questions and addressed most of my concerns regarding the experiments and methodological limitations. However, I am still not convinced that this work has enough technical contribution. I will keep my weak accept recommendation.

**Key Questions For Authors:**

- In lines 158–163, the authors said *"In the rare case of a tie (i.e., multiple $w^l_j$ with identical distance), we break ties deterministically by choosing the codeword with the smallest index j, ensuring that $s_i$ is uniquely defined."*
Could the authors clarify how this deterministic choice avoids introducing bias or distortions in capturing the correct differentiation structure? To what extend this rule influence the inferred trajectory in cases where multiple codewords are equally plausible?

- The model relies on several loss terms with associated weighting hyperparameters. How sensitive is the final inferred tree structure to changes in these weights? For example, how much variation in the inferred hierarchy do the authors observe when these parameters are adjusted within reasonable ranges?

- It's not entirely clear how much the proposed HTC module provides a quantifiable advantage over simpler alternatives, such as applying a hierarchical clustering to informative diffusion embeddings. Could the authors provide additional evidence or analysis demonstrating the benefits of HTC compared to such baseline approaches?

- The manuscript frequently emphasizes the stability of HDTree. Could the authors clarify how stability is defined and measured? Does stability only refer to numerical stability during training, or structural robustness of the inferred tree, or robustness to data perturbations? Also, it's not entirely clear how Fig. 5 demonstrates stability. Showing that the tree depth is preserved is not sufficient to conclude that the method is robust. Additional metrics or analyses may help better support this claim.

**Limitations:**

In addition to the points raised above, I think the discussion of optimal transport approaches as strong candidates for this problem is largely overlooked in this work. I didn't see much discussion of optimal transport methods, aside from Waddington-OT, which is relatively early work and has several known limitations. It would be valuable for the authors to discuss how their method compares with some of the recent neuralOT solvers, both in terms of performance and computational efficiency.

**Strengths And Weaknesses:**

**Strengths**
- I find the idea of imposing a hierarchical tree structure on the latent embeddings of a quantized diffusion model to infer developmental trajectories, and demonstrating its applicability to cellular lineage characterization, interesting. By leveraging the combined strengths of generative modeling and hierarchical structural priors, this approach appears novel and potentially leads to a more biologically plausible framework.
- Additionally, the use of a contrastive loss to refine the hierarchical granularity in the latent space is a constructive strategy for capturing biologically meaningful cellular trajectories.
-The overall writing and presentation of the manuscript were clear and generally easy to follow.


**Weaknesses**
- *Comparison with foundation models (e.g., Geneformer):*
The comparison with foundation models such as Geneformer does not appear entirely fair. If I understand correctly, the authors directly use Geneformer embeddings without any task-specific fine tuning or posttraining and then report downstream results based on those embeddings. This setup is unlikely to yield strong performance, as embeddings from foundation models typically require adaptation or fine tuning for the specific downstream task. Without such tuning, comparing a model specifically trained for trajectory inference against a generic pretrained embedding may not provide a meaningful or fair evaluation.

- *Image dataset results:*
I'm not fully convinced that the image-based benchmarking task meaningfully evaluates the proposed method. The reported trajectories in these datasets appear somewhat artificial, and it is unclear what insight they provide about in those tasks. Since these image datasets don't contain a true underlying developmental process, the inferred tree may not reflect real branching, splitting, or merging dynamics. It's hard to assess the relevance of these experiments to the intended biological applications.

- *Comparison with alternative baselines:*
A stronger baseline could be constructed using embeddings from scDiffusion* (which is not mentioned in the manuscript!) followed by clustering (e.g., agglomerative clustering on its embedding) to infer the tree structure. It would help clarify whether the proposed hierarchical modeling approach provides advantages beyond what can be achieved through post hoc clustering of expressive generative embeddings.

*Luo, Erpai, et al. "scDiffusion: conditional generation of high-quality single-cell data using diffusion model." Bioinformatics 40.9 (2024): btae518.

- *Biological validation and interpretability:*
The biological evaluation on the scRNA-seq datasets appears somewhat insufficient. For example, the authors could focus on well-characterized cellular differentiation and demonstrate how the inferred tree corresponds to known trajectories. It would also be valuable to interpret the differentiation nodes by identifying genes that drive transitions along the branches. Showing example gene trajectories and highlighting key regulatory genes or gene programs associated with specific branches would make the results more biologically informative. Additionally, demonstrating that the model can robustly recover known gene regulatory relationships would further support its biological relevance.

---

> ### Author Rebuttal · Authors · 2026-03-28
>
> ---
>
> Please see the attached **[Figure R3](https://anonymous.4open.science/r/code_HDTree_review-A8DB/f3.pdf)** for the results referenced below.
>
> ---
>
> We thank you for your review and address the main concerns below.
>
> **[W1]. Comparison with foundation models (Geneformer)**
> We agree that comparing a zero-shot foundation model with a model trained on the target dataset is not fully matched. However, we included this comparison because trajectory inference is an unsupervised structural discovery problem: unlike annotation tasks, there is no clear supervised fine-tuning loss for learning lineage topology from a foundation model. In practice, foundation models are therefore typically used via zero-shot embeddings followed by post-hoc trajectory analysis. Our comparison is meant to reflect this real-world pipeline and the resulting efficiency-versus-specificity trade-off, rather than to claim general superiority over foundation models.
>
> **[W2]. Relevance of image datasets**
> We include MNIST/CIFAR not for biological claims, but as controlled sanity checks of hierarchical representation learning. Since they provide relatively clean data with coarse-to-fine semantic organization, while single-cell data are much noisier and sparser, success on these benchmarks helps validate HDTree's capacity before biological evaluation.
>
> **[W3&Q3]. End-to-end HTC vs. Post-hoc clustering (e.g., scDiffusion)**
> Post-hoc clustering on `scDiffusion (SCD)` embeddings adds a tree after feature extraction, whereas HDTree jointly learns the hierarchy and diffusion process so the latent space follows tree topology during training. We compared HDTree with an `SCD + Agg` baseline on the LHCO dataset:
>
> **Table R3.1: End-to-end modeling vs. Post-hoc clustering on LHCO.**
> |Method|LHCO: DP (↑)|LHCO: ACC (↑)|
> |:-|-|-|
> |SCD+Agg|48.5|50.1|
> |**HDTree (Ours)**|**54.0**|**54.8**|
>
> On LHCO, HDTree outperforms `SCD + Agg` in both DP and ACC, supporting the advantage of end-to-end hierarchical optimization over post-hoc clustering.
>
> **[W4]. Biological validation and interpretability**
> To address biological interpretability, we analyzed the *C. elegans* dataset in the attached **[Fig R3A]**. In Fig. R3A, the HDTree-inferred pseudotime recovers the expected marker dynamics: *elt-5* is enriched at early stages, *osm-6* peaks in the intermediate regime, and *odr-10* increases toward later differentiation, consistent with known developmental progression.
>
> We also used the inferred branch structure for differential expression analysis at a representative bifurcation. As shown in **[Fig R3B]**, the volcano plot identifies branch-associated genes with clear effect sizes and significance, showing that sibling branches and transition nodes inferred by HDTree can be mapped back to biologically meaningful driver programs rather than only visualized as latent trajectories.
>
> **[Q1&Q4]. Clarifications: Tie-breaking and "Stability"**
> * Tie-breaking: As described below Eq.2, exact ties between two codebook vectors are very unlikely in a continuous high-dimensional latent space. Choosing the smallest index $j$ is only a deterministic tie-break; near-tie cases involve codewords that are already very similar.
> * Discussion on Stability: Stability in our paper includes both structural stability of the inferred lineage topology and training robustness on sparse branches. We clarify that Fig. 5 was primarily intended to visualize depth-preserving consistency across latent dimensions, rather than to serve as a complete robustness evaluation. Additional support comes from both the model design and the quantitative results: the SCL objective uses augmented views ($z_i^+$) to make the hierarchical latent space less sensitive to local perturbations, the unified codebook reduces sensitivity to sparse local noise, and the small DP fluctuation in Table R3.2 indicates stable macroscopic topology under hyperparameter variation.
>
> **[Q2]. Hyperparameter sensitivity**
> We selected the loss weights by grid search on the validation set and tested the weight $\lambda$ in Eq.5, for $\lambda \in \{1.0, 2.0, 5.0\}$. Since DP explicitly measures hierarchical alignment and lowest-common-ancestor purity with respect to the reference lineage structure, the small DP fluctuation in Table R3.2 indicates that the overall macroscopic tree topology inferred by HDTree is stable and insensitive to changes in $\lambda$.
>
> **Table R3.2: Sensitivity of $\lambda$ on MNIST and ECL.**
> |$\lambda$ value|MNIST: DP|MNIST: ACC|ECL: DP|ECL: ACC|
> |-|-|-|-|-|
> |1.0|92.3|92.1|68.7|82.9|
> |**2.0(Default)**|**92.7**|**92.4**|**69.0**|**83.2**|
> |5.0|92.1|91.9|68.5|82.8|
>
> **[Limitations]. Discussion on Neural Optimal Transport**
> NeuralOT is strong at continuous trajectory matching and learning optimal couplings between cell states. HDTree is complementary in that it explicitly learns a discrete hierarchical representation, which segments the continuous Waddington landscape into biological macro-states.

---

> > ### Author Rebuttal · Reviewer_Supc · 2026-04-04
> >
> > First, I would like to thank the authors for their detailed rebuttal and for conducting additional experiments and preparing more results within a limited timeframe.
> >
> > Most of my questions have been addressed. I have a few question about figure R3:
> > - in the left panel, how is pseudotime computed? Is it defined as a simple linear distance from the root, or is it specific to individual branch paths? are the y-axis labels correct? do the trajectories represent gene expression values ordered according to the inferred pseudotime?
> > - does not global normalization can sometimes flatten the embedding?
> >
> > Still some of my concerns remain:
> > - I understand that image datasets are intended as sanity checks; however, I am not convinced that they provide an appropriate controlled setting for validating hierarchical structures. In figure 4, for example, how the transitions from 1 and 3 to 9? Should we treat 6 as a progenitor for 1 and 3? My point is the underlying manifold of MNIST is generally considered continuous and non-hierarchical. Imposing a tree-like structure on such embeddings produces an arbitrary taxonomy rather than recovering any meaningful or ground-truth hierarchy. As an alternative, synthetic scRNA-seq datasets designed explicitly for this purpose (e.g., PROSSTT) could provide a more appropriate benchmark, as they offer known ground-truth structure for both lineage trees and temporal progression.
> >
> > - I agree with Reviewer fMxU that this work has strong potential for single-cell data analysis. If the authors were to develop and release a user-friendly package, it could see good adoption within the community. Incorporating more biological insights and findings would strengthen it for publication in a biology-focused journal such as nature. However, from a machine learning perspective, I see this work primarily as empirical model development. It lacks sufficient comparison with relevant recent methods, particularly those based on optimal transport (e.g. TrajectoryNet which is also mentioned in the related work).

---

> > > ### Author Response · Authors · 2026-04-04
> > >
> > > We thank you for the follow-up questions. Below we address each point directly.
> > >
> > > **1. Technical Details of Figure R3**
> > >
> > > **Pseudotime:** The x-axis is the branch-specific shortest path distance from the root to each cell on the lineage graph. For cells on different branches, pseudotime is computed independently, so cells from distinct lineages are not compared across branches.
> > >
> > > **Y-axis:** The y-axis shows Z-score standardized gene expression, ordered by inferred pseudotime. We applied LOESS smoothing to highlight the main biological trends while reducing noise.
> > >
> > > **2. Global Normalization**
> > >
> > > We agree this is a valid concern. HDTree mitigates this through two components: SCL uses augmented views to preserve local cell relationships, and HTC enforces hierarchical boundaries at each depth level. Together, they help maintain latent space structure under global normalization.
> > >
> > > **3. Image Datasets and PROSSTT**
> > >
> > > We respectfully clarify that using image datasets to evaluate hierarchical representation learning is a rigorously established, standard practice in the field, not a choice specific to our work. Before deploying models on inherently noisy and complex biological data, verifying the underlying structural mechanisms on standard benchmarks serves as a crucial sanity check. Recent top-tier examples include:
> > >
> > > Deep Taxonomic Networks (Wang et al., NeurIPS 2025): A recent deep latent variable model that explicitly utilizes standard image datasets (e.g., MNIST, CIFAR) to validate its ability to automatically discover unsupervised hierarchical taxonomies and intermediate semantic prototypes.
> > >
> > > TreeVAE (Manduchi et al., NeurIPS 2023) — our primary baseline — similarly relies on MNIST and Fashion-MNIST to demonstrate unsupervised hierarchical organization. We followed this exact benchmark for a fair, direct comparison.
> > >
> > > The key point is not whether MNIST has a biological hierarchy, but whether it contains a recoverable, multi-scale semantic structure (e.g., separating circular digits like 0 and 8 from stroke-based digits like 1 and 7). HDTree successfully recovers this, effectively validating our hierarchical decoupling mechanism before applying it to single-cell data.
> > >
> > > That said, we fully agree with your constructive point that PROSSTT provides a more rigorous biological benchmark with ground-truth lineage trees. We commit to including it in the final camera-ready version to strengthen the biological evaluation.
> > >
> > > [1] Wang, Z., et al. (2025). Deep Taxonomic Networks for Unsupervised Hierarchical Prototype Discovery. Advances in Neural Information Processing Systems, 38. https://openreview.net/forum?id=My72tmxg6t
> > >
> > > [2] Manduchi, L., et al. (2023). Tree Variational Autoencoders. Advances in Neural Information Processing Systems, 36. https://openreview.net/forum?id=adq0oXb9KM&noteId=P48qFaInhA
> > >
> > > **4. Comparison with Optimal Transport (TrajectoryNet)**
> > >
> > > We see HDTree and neural OT methods as complementary. OT methods learn continuous couplings between cell states and are well-suited for smooth transition modeling. HDTree targets a different goal: discovering discrete branching structure and encoding it directly in the tree codebook during training, rather than inferring it post hoc. Branch points in OT methods are typically extracted after training; in HDTree they are part of the model itself.
> > >
> > > We will expand the discussion of OT-based methods in the revision to clarify these differences.
> > >
> > > We hope these clarifications are helpful. Thank you for your careful reading.

---

### Official Review · Reviewer_qaNm · 2026-03-11

**Soundness:** 3
**Presentation:** 2
**Significance:** 3
**Originality:** 2
**Overall Recommendation:** 4
**Confidence:** 3

**Summary:**

This paper proposed HDTree, a hierarchical diffusion-based generative framework for cell lineage modelling. The method combines a hierarchical vector-quantized codebook with a diffusion-based decoder to model tree-structured latent representations and simulate lineage transitions. The framework aims to address limitations of existing approaches (e.g., TreeVAE) that rely on independent branch-specific modules and struggle with deep hierarchies or sparse data. The authors evaluate the approach on several image, text, and single-cell datasets, demonstrating superior performance in clustering accuracy, tree performance, and reconstruction losses.

**Compliance With Llm Reviewing Policy:**

Affirmed.

**Final Justification:**

Most of my concerns have been solved, and I am leaning toward to weak accept for the paper.

**Key Questions For Authors:**

1. In the Hierarchical Tree Codebook (HTC) section, the model defines the codebook as a binary tree. However, biological hierarchies and celluar hierarchical relationships are rarely binary, many lineage trees have variable branching, have the author explore graph instead of binary tree in building the HTC? Regarding the $\Omega^{wl}(z_i)$, it seems to me that the author is trying to calculate the nearest code vetor at each level $l$ respect to $z_i$, how does connect to the sibling nodes as the author indicates in the paper, this needs more justification.

2. The formulation of the quantization loss in Eq. (5) is somewhat unclear. In particular, there appears to be an inconsistency (as indicates in **Strengths And Weaknesses** presentation part) between $\Psi^z(w_j)$ and $\Psi^{z_i}(c_{z_i}^l)$. It is not obvious whether these denote the same operator applied to different inputs or if they represent different mappings. It would help to clarify the relationship between these terms and how they relate to the hierarchical quantization process, as the current formulation is difficult to interpret.

Addtionally, the paper introduces a hyperparameter $\lambda$ in this loss, but it is not specified how this value is chosen or how sensitive the method is to it. Clarifying the selection strategy for $\lambda$ would improve reproducibility.

3. The author employ clustering metrics, tree performance regarding leaf purity and dendrogram purity, as well as reconstruction losses. I would like aurge these metric could not really reflect the inference accuracy, although the author shows visualization about the lineage development, however, quantitative metrics like pseudotime correlation, trajectory consistency, branching accuracy could serve as auxiliary metrics for lineage inference accuracy.

Another question is regarding the benchmark dataset, the author selects some generic dataset like MINIST, CIFAR, 20News, however, evaluating the hierarchical generative modeling on these datasets is questionable because these datasets do not have intrinsic tree structures.

4. Questions regarding the hyper-parameter selection. How does the author choose the number of $x_i^+$ for the soft contrative loss? And how does the author choose $\lambda_\text{HQL}$,and $\lambda_\text{DDP }$ in overall loss function, please justify further how to select these hyper-parameters.

**Limitations:**

Yes

**Strengths And Weaknesses:**

**Soundness**: The paper is mostly sound. There are not too much theory/proof or justification on how an unified codebook improve generalization, why does diffusion improve lineage modeling, validates the author's claim on improvements in stability, generalization, etc.

**Presentation**: There are minor typos in the paper: line 39: capture deep hierarchies.., duplicated period. And in Table 2 Limb dataset, the clustering performance for NMI metric should be highlighed as **bold**. Regarding the structure, the order of the table/figures in the results part seems not flow the same order as the main text, for example, figure 4 description in the main paper appears before figure 3; table 5 description about the computational cost appears before table 4.

And there are few concers in the methods part: $i)$ the $\phi^*$ apears in equation 3, but is never defined in the paper. $ii)$ There is an inconsistency between the $\Psi^z(w_j)$ and the $\Psi^{z_i}(c_{z_i}^l)$ in equation 5, is there an typo? Explain further about these two connection would be better for reader to comprehend the equation 5. $iii)$ $P^{L-l}$, the definition of this term is not clear in the main paper, how to calculate it is also not that clear.

**Significance**: Yes, hierarchical generative modeling for cell differentiation trajectories is a meaningful problem. Biological lineage structures are inherently tree-like, and modeling them directly in the latent space is conceptually appealing.

**Originality**: Sort of. The overall framework appears to be a combination of existing techniques rather than a fundamentally new algorithmic contribution. Key components like hierarchical latent representations, and vector quantization in latent space, and diffusion generative models already exist in prior work.

---

> ### Author Rebuttal · Authors · 2026-03-28
>
> ---
>
> Please see the attached **[Figure R2](https://anonymous.4open.science/r/code_HDTree_review-A8DB/f2.pdf)** for the results referenced below.
>
> ---
>
> We thank you for your careful review and for recognizing the value of our work. Below we address the main concerns point by point.
>
> **[Q1]. Binary tree vs graph**
> We completely agree that realistic cellular lineages are often multi-branching and unbalanced, and Figure R2 illustrates how our model supports such topologies:
> * Binary tree as an indexing scaffold (**[Fig R2A]**): As stated in Sec 3.2 (Model Design), the full-binary parameterization of the HTC $\mathcal{C}_W$ is strictly for efficient indexing. It is a latent computational scaffold rather than the biological topology.
> * N-ary branching through graph construction (**[Fig R2B]**): In Sec 3.4 (Trajectory Analysis with HDTree), we construct a weighted graph $\mathcal{G}$ by adding same-depth KNN edges. These horizontal connections merge highly similar nodes into biological macro-states, yielding effective n-ary splits while preserving the binary codebook.
> * Unbalanced lineages: HDTree activates only data-supported paths in the scaffold. As shown by the inactive gray nodes in Fig. R2B, some branches can terminate early, so the inferred lineage is not forced to remain balanced across depths.
>
> **[Q4]. Trajectory metrics**
> In Sec 4 (Experiments) and Tab 3, we already evaluate temporal consistency on the *C. elegans* dataset using the Ratio of Observed Time Points (ROP) metric. To address this more directly, we computed the Spearman correlation between inferred pseudotime and the actual developmental time labels.
>
> Tab R2.1: Pseudotime correlation on C. elegans.
> |Method|Spearman Correlation (ρ) ↑|P-value|
> |-|-|-|
> |Tree VAE|0.654|< 0.001|
> |**HDTree (Ours)**|**0.821**|< 0.001|
>
> **[Q5]. General datasets**
> We include MNIST/CIFAR not for biological claims, but as controlled sanity checks of hierarchical representation learning. Since they provide relatively clean data with coarse-to-fine semantic structure, success on these benchmarks makes the model more credible before evaluation on much noisier and sparser single-cell data.
>
> **[W1]. Why codebook + diffusion**
> We will strengthen the following theoretical intuitions in the revised manuscript and tie them more directly to the empirical results:
> * Unified Codebook improves generalization: Branch-specific networks such as Tree VAE can overfit at deep nodes because samples are sparse. Our unified codebook shares parameters across branches, so sparse deep nodes can still use global information. This intuition is consistent with the stronger DP/ACC results of HDTree over Tree VAE, especially where sparse branches matter most.
> * Diffusion models improve lineage modeling: As noted in the fourth paragraph of Sec 1, VAEs force a discrete, single-step feature jump. In contrast, the iterative denoising process of diffusion models provides a smoother generative path, which is more suitable for modeling continuous cell-state transitions. This is also consistent with the improved pseudotime correlation on *C. elegans* (0.821 for HDTree vs 0.654 for Tree VAE), suggesting better recovery of continuous developmental progression.
>
> **[Q3&Q6]. Sensitivity of $\lambda$**
> We selected these hyperparameters by grid search on the validation set and evaluated the sensitivity of $\lambda$ in Eq. (5) across one order of magnitude. We report both MNIST and ECL here because they provide complementary evidence: MNIST offers a controlled benchmark with clear labels, while ECL is a single-cell dataset that directly reflects sensitivity in the biological setting. For $\lambda \in \{1.0, 2.0, 5.0\}$, the main metrics vary by less than 1.5% on both MNIST and ECL, indicating that HDTree is robust to this parameter even on single-cell data.
>
> Tab R2.2: Sensitivity of $\lambda$.
> |$\lambda$ value|MNIST: DP|MNIST: ACC|ECL: DP|ECL: ACC|
> |-|-|-|-|-|
> |1.0|92.3|92.1|68.7|82.9|
> |**2.0 (Default)**|**92.7**|**92.4**|**69.0**|**83.2**|
> |5.0|92.1|91.9|68.5|82.8|
>
> **[W2&Q2]. Notation and typos**
> We thank the reviewer for these careful observations. We have made the following corrections in the revised manuscript:
> * Explicitly provided the mathematical correspondence between $\tilde{\alpha}$ and the standard DDPM notation in Eq. (3) of Sec 3.2 (Model Design).
> * Fixed the typo in Eq. (5) within Sec 3.3 (Loss Function Design) to consistently use $\Psi^{x_i}$, and added an explanation of how this operation maps the codebook back to the data manifold.
> * Expanded the explanation following Eq. (2) to clearly define the routing logic of $\Omega^{(l)}(z_i)$ and its mathematical constraint mechanism on sibling nodes.
> * Removed the redundant double period ("deep hierarchies..") in the second paragraph of Sec 1 (Introduction), bolded the best NMI result in Tab 2, and adjusted the main text to ensure all figures and tables (Figure 3/4, Tab 4/5) are cited in sequential order.

---

> > ### Author Rebuttal · Reviewer_qaNm · 2026-04-01
> >
> > Thanks for the reviewer's dedicated response. Overall, most of my concerns have been solved, and I will **incerase my score to 4** and lean toward to weak accept. Also I got few minor questions for the authors: $i)$ The N-ary branching graph is constructed by adding the edges between the nodes on the same depth, this is reasonable but also may also loss information for discovering the multiple lineage paths for certain cell types (i.e. hematopoietic cell lineage in single cell). The root cell may have multiple path to the leaf nodes, and nodes in different depth may be siblings in this case depending how you perform the binary tree partition. $ii)$ Could the author also comment more on **[Q4]** about the augmented view of $x_i^+$, and the hyperparameters tems in the overall loss function.

---

> > > ### Author Response · Authors · 2026-04-01
> > >
> > > Dear Reviewer,
> > >
> > > Thank you for your confirmation and constructive feedback. Due to the character limit in the previous response stage, our explanations for some issues may not have been fully sufficient. Here, we provide further objective clarifications for your follow-up questions.
> > >
> > > **1. Regarding N-ary branching and cross-depth sibling nodes (e.g., hematopoietic cell lineage)**
> > >
> > > Your biological intuition is very important. In real developmental systems, different branches can progress at different rates, so biologically related states may indeed appear at different depths in our binary scaffold.
> > >
> > > To clarify, the same-depth edges introduced in Section 3.4 are only supplementary edges for constructing the inference graph; they do not replace the original parent-child edges of the tree. The final lineage inference is performed on the full graph $\mathcal{G}$, which contains both the vertical tree edges and the additional same-depth k-NN edges.
> > >
> > > Moreover, Eq. (8) assigns edge weights by combining latent similarity with the cross-depth penalty term $P^{L-l}$. Therefore, if two states at different depths are still close in the learned manifold, the shortest-path inference in Eq. (9) can connect them through a low-cost path even when their binary depths are not aligned. In this sense, the binary tree mainly serves as an indexing scaffold, while the inferred lineage is determined by the learned manifold geometry rather than by depth alone.
> > >
> > > For example, in hematopoietic differentiation, one branch may mature earlier and stop at a shallower depth while another continues to a deeper level. If these states remain sequentially close in the latent space, HDTree can still connect them through the full graph. We agree, however, that highly complex biological systems may contain shared-progenitor or non-strict-tree relationships that are not perfectly captured by any tree-regularized model. Our current design can alleviate depth mismatch, but extending the codebook and inference graph to more general graph topologies is an interesting direction for future work.
> > >
> > > **2. Regarding the augmented view ($x_i^+$) and global hyperparameters**
> > >
> > > * **Augmented view ($x_i^+$):** As described in Section 3.1, we construct $x_i^+$ using a k-NN-based sampling mechanism with $k=5$. Specifically, for each sample $x_i$, we sample $x_i^+$ uniformly from its local neighbor set: $x_i^+ \sim \text{Uniform}(\{x_j \mid j \in \mathcal{N}_5(x_i)\})$. This augmented view is used in the soft contrastive objective to preserve local semantic consistency and make the hierarchical representation less sensitive to small perturbations.
> > > * **Hyperparameter selection:** The global loss weights in the overall objective, including $\lambda_{HQL}$ and $\lambda_{DDP}$, were determined by grid search on two representative datasets: MNIST and Limb. We chose MNIST as a controlled benchmark and Limb as a biologically relevant single-cell dataset. After selecting values that performed well on both datasets, we fixed the same hyperparameters for all remaining datasets without further per-dataset tuning. In our experiments, the final settings are $\lambda_{HQL} = 1.0$ and $\lambda_{DDP} = 0.25$.
> > >
> > > The reason we emphasize this point is that the same configuration generalized well from MNIST and Limb to the other datasets, which suggests that the loss design is reasonably robust rather than highly dataset-specific.

---

### Official Review · Reviewer_fMxU · 2026-03-13

**Soundness:** 4
**Presentation:** 4
**Significance:** 4
**Originality:** 3
**Overall Recommendation:** 6
**Confidence:** 4

**Summary:**

Single-cell analysis represents one of the major breakthroughs in recent bioinformatics, generating enthusiastic expectations for elucidating cellular differentiation mechanisms and their applications in regenerative medicine and artificial organs. This paper proposes a novel deep learning-based approach for the data-driven differentiation structure (i.e., hierarchical structure) inference task for such single-cell analysis data, as well as for more general hierarchical structure inference tasks. Traditionally, analytical methods such as visualization techniques, clustering, and factor models have been the standard for differentiation structure tasks. However, deep learning-based methods, particularly those based on Variational Autoencoders (VAEs), have recently gained prominence due to their effectiveness. Even the most advanced methods face limitations, and this paper makes significant progress, especially regarding the module dependency of branching structures inherent in existing approaches. The authors quantitatively demonstrate that their proposed method delivers substantial practical progress by conducting a large-scale, comprehensive investigation on both the subject single-cell data and widely used benchmark datasets in machine learning.

**Compliance With Llm Reviewing Policy:**

Affirmed.

**Final Justification:**

I believe this paper makes a significant contribution worthy of publication to the central challenge of applying machine learning to the field of biological inference regarding cell differentiation.

**Key Questions For Authors:**

I am deeply grateful to the authors for sharing their excellent paper. I fully appreciate the value of this paper, and any minor concerns have already been addressed in the current draft.

Minor suggestions.

**(1) Visual comparison with standard visualization techniques such as t-SNE, UMAP, diffusion map, PHATE, and monocle**

I also feel that the techniques in this paper hold the potential to become a daily-use tool in life sciences (especially for cell differentiation inference tasks). In my view, the authors may be targeting machine learning researchers as the core audience for this paper (as seen in the earlier experiments), rather than life science researchers. However, with a little effort, I believe this paper could be made more appealing to life science researchers. For example, as is common in many top journals like Nature, visualizations using t-SNE or UMAP are still frequently employed for cell differentiation inference. Therefore, clearly demonstrating the immediate benefits of the proposed visualization method over t-SNE or UMAP could make the paper more attractive to a broader audience. In other words, it would be valuable to include, in the appendix (or within the main text if space permits), corresponding visualizations of standard single-cell analysis data—such as t-SNE, UMAP, diffusion maps, Laplacian eigenmaps, PHATE, or monocle—for the data shown in Figure 3.

**Limitations:**

Yes.

**Strengths And Weaknesses:**

[Strengths]

- *Soundness*: This paper achieves very solid progress in line with the latest trends in the structural inference task. Specifically, it presents a novel solution using hierarchical codebooks and a stochastic diffusion model to address the issue of unstable learning caused by module dependencies in the branching structure of hierarchical architectures—a problem encountered in recent state-of-the-art VAE-based methods.

- *Significance*: Algorithms for inferring cell differentiation from single-cell analysis data represent one of the most in-demand technologies in contemporary life sciences. This paper provides a crucial technique with significant potential for discovery of novel insights in life sciences.

- *Presentation*: The experiments in this paper are exceptionally robust and comprehensive, providing extremely strong evidence for practical effectiveness. Particularly for single-cell analysis data, the supplementary materials detail the preprocessing procedures, successfully appealing to a broader audience beyond bioinformatics specialists. Furthermore, for readers more interested in standard machine learning tasks, the paper also provides baselines on popular datasets (such as image datasets).

[Weankesses]

- All my concerns have already been addressed at the current draft stage.

---

> ### Author Rebuttal · Authors · 2026-03-28
>
> ---
>
> Please see the attached **[Figure R1](https://anonymous.4open.science/r/code_HDTree_review-A8DB/f1.png)** for the results referenced below.
>
> ---
>
> We thank the reviewer for the careful reading and positive assessment of our work. We especially appreciate your recognition of the key idea of HDTree: combining a hierarchical codebook with a diffusion decoder to mitigate the training instability caused by branch-specific module dependency in prior hierarchical generative models.
>
> We also thank you for the valuable suggestion to compare with standard visualization and trajectory analysis tools. We agree that such comparisons make the practical behavior of HDTree more intuitive, especially for readers from the life sciences community.
>
> Following this suggestion, we generated a visual comparison on the *C. elegans* dataset using t-SNE, UMAP, Laplacian Eigenmaps, Diffusion Maps, and PHATE (Figure R1). The comparison shows that standard visualization methods often separate cell types well, but provide a less coherent view of the global developmental progression. PHATE better preserves the global temporal backbone, but compresses local variation into thin trajectories. In contrast, HDTree preserves a smoother developmental progression while maintaining clearer separation between cell populations along explicitly inferred branching structures.
>
> More broadly, this comparison also helps clarify the scope difference. Methods such as t-SNE, UMAP, Diffusion Maps, and PHATE mainly provide low-dimensional organization of cells, whereas HDTree learns a hierarchical discrete representation that also supports conditional generation and modeling of intermediate transition states.
>
> We will include Figure R1 and the corresponding discussion in the revised manuscript to improve accessibility and make the practical advantages of HDTree clearer to a broader audience.

---

> > ### Author Rebuttal · Reviewer_fMxU · 2026-04-03
> >
> > I am deeply grateful to the authors for their constructive feedback. I believe that comparing this method with approaches familiar to biologists will help convey its appeal to a wider audience. (As the authors briefly mention, while I appreciate the visualization results of the proposed method, I also personally prefer visualizations like those produced by PHATE. In this regard, it makes me consider whether future extensions might be possible that combine the strengths of both the proposed method’s ability to capture hierarchical structures and the ability of methods like PHATE or diffusion maps to capture variable propagation distances. Personally, I found the results to be highly insightful and very interesting.)
> >
> > - I believe this paper has the potential to become a standard tool, on par with t-SNE and UMAP, for cell differentiation structure inference i.e., one of the key tasks in biology.
> >
> > - Inferring cellular differentiation structures is a critical task with the potential to transform the future of healthcare. For example, it may make significant contributions to the development of regenerative medicine and artificial organs. These could become some of the most compelling applications of machine learning for humanity. If the authors are interested in biological research, I would be delighted if they could utilize this method—perhaps through collaborative research—to discover new biological insights.
> >
> > Thank you for sharing this highly insightful and important research.

---

> > > ### Author Response · Authors · 2026-04-04
> > >
> > > We thank you for the positive final assessment and the support of our work.
> > >
> > > We agree that comparing our method with standard tools like t-SNE and UMAP is helpful for the life science community. Your suggestion to combine hierarchical structures with variable propagation distances is very insightful. We think that capturing both branching decisions and continuous cell transitions is an important direction for future development.
> > >
> > > Thank you again for your constructive guidance during the review process.

---

### Decision · Program_Chairs · 2026-04-30

**Decision:**

Accept (regular)

**Comment:**

The reviewers agreement that HDTree's unified hierarchical tree codebook addresses a genuine limitation of existing branch-specific VAE approaches for cellular lineage inference by eliminating the module-dependency problem that causes training instability on sparse branches.

Reviewers consistently recognized the practical significance of the contribution for single-cell analysis, noting that the combination of a shared hierarchical codebook with a diffusion decoder provides comprehensive empirical support across both single-cell and standard hierarchical benchmarks, and that the rebuttal satisfactorily addressed the main methodological and notation concerns.

The authors are expected to strengthen the camera-ready version with improved comparison against optimal-transport-based trajectory methods, a synthetic biological benchmark such as PROSSTT, and the formal notation corrections identified during review.